# Regulation of *Myc* transcription by an enhancer cluster dedicated to pluripotency and early embryonic expression

Lin Li-Bao[1,10], Covadonga Díaz-Díaz[1,2], Morena Raiola[1], Rocío Sierra[1,2], Susana Temiño[1,2], Francisco J. Moya[3], Sandra Rodriguez-Perales [3], Elisa Santos[4], Giovanna Giovinazzo [2,4], Tore Bleckwehl[5,6], Álvaro Rada-Iglesias [7], Francois Spitz [8,9] & Miguel Torres [1,2] ✉

MYC plays various roles in pluripotent stem cells, including the promotion of somatic cell reprogramming to pluripotency, the regulation of cell competition and the control of embryonic diapause. However, how *Myc* expression is regulated in this context remains unknown. The *Myc* gene lies within a ~3-megabase gene desert with multiple cis-regulatory elements. Here we use genomic rearrangements, transgenesis and targeted mutation to analyse *Myc* regulation in early mouse embryos and pluripotent stem cells. We identify a topologically-associated region that homes enhancers dedicated to *Myc* transcriptional regulation in stem cells of the pre-implantation and early post-implantation embryo. Within this region, we identify elements exclusively dedicated to *Myc* regulation in pluripotent cells, with distinct enhancers that sequentially activate during naive and formative pluripotency. Deletion of pluripotency-specific enhancers dampens embryonic stem cell competitive ability. These results identify a topologically defined enhancer cluster dedicated to early embryonic expression and uncover a modular mechanism for the regulation of *Myc* expression in different states of pluripotency.

MYC is a transcription factor (TF) that regulates a large number of downstream genes, most of them involved in cellular growth and proliferation. MYC is one of the four factors able to reprogram somatic cells to pluripotency[1]. Pluripotency is the capacity of embryonic cells to self-renew keeping the potential to differentiate into the three embryonic germ layers. In the mouse embryo, pluripotency appears first in the epiblast of the blastocysts before implantation around embryonic day 3.5 (E3.5). Pluripotency is then maintained in the

epiblast through implantation until late gastrulation (E7.5). Although a core pluripotency TF regulatory network can be recognized in epiblast cells throughout this period, cells transit through different pluripotency states that reflect their preparation for differentiation. Two extreme states have been defined; "naive" and "primed" pluripotency, which show differences in gene expression, epigenetic landscape, signal transduction, and metabolic profile[2–4]. The naive state is characterized by a generalized hypomethylated "open" chromatin state,

[1]Cardiovascular Regeneration Program, Centro Nacional de Investigaciones Cardiovasculares (CNIC), Madrid, Spain. [2]Centro de Investigación Biomédica en Red de Enfermedades Cardiovasculares (CIBERCV), Madrid, Spain. [3]Molecular Cytogenetics and Genome Editing Unit, Human Cancer Genetics Program, Centro Nacional de Investigaciones Oncológicas (CNIO), Madrid, Spain. [4]Pluripotent Cell Technology Unit, Centro Nacional de Investigaciones Cardiovasculares, CNIC, Madrid, Spain. [5]Center for Molecular Medicine Cologne (CMMC), University of Cologne, Cologne, Germany. [6]Institute of Experimental Medicine and Systems Biology, RWTH Aachen University, Aachen, Germany. [7]Institute of Biomedicine and Biotechnology of Cantabria (IBBTEC), CSIC/ University of Cantabria, Santander, Spain. [8]Department of Human Genetics, The University of Chicago, Chicago, IL, USA. [9]Developmental Biology Unit, European Molecular Biology Laboratory, Heidelberg, Germany. [10]Present address: Centro Andaluz de Biología del Desarrollo (CABD), Sevilla, Spain. ✉e-mail: mtorres@cnic.es

including both X-chromosomes in female embryos and the absence of any germ layer marker. The naive state is therefore considered as the unbiased pluripotent state and is established around the time epiblast segregates from primitive endoderm. After implantation, epiblast cells undergo a process of "formative pluripotency"[5] which consist of a progressive molecular re-wiring towards the primed state, attained around the start of gastrulation[2,3,6]. Primed epiblast cells have already established X chromosome inactivation and have gained extensive methylation marks[2]. Pluripotent cells in culture constitute a powerful tool to study the transition between pluripotency states. Mouse Embryonic Stem Cells (mESCs) can be derived from the E3.5 epiblast, maintained in a naive state, and be differentiated to epiblast-Like Cells (EpiLCs), which represent the formative pluripotency state, and epiblast Stem Cells (EpiSCs), which represent the primed pluripotency state. Each of these in vitro states can be stably or transiently induced in culture by using specific conditions[5]. Specifically, the EpiLCs state can be transiently obtained in vitro using conditions that lead to the EpiSC state after a few days in culture[7].

*Myc* is expressed throughout all pluripotency states and is repressed in primed cells close to differentiation[8–10]. Throughout the pluripotency states, *Myc* shows very heterogeneous levels of expression between individual cells, which results in spontaneous cell competition in mESC cultures and epiblast cells[8,10]. Cell competition is a homeostatic mechanism whereby equivalent neighboring cells compare their fitness eventually leading to the elimination of the less fit population (loser) and its replacement by the surviving population (winner). In the context of pluripotent cells, the elimination of MYC-low cells has been proposed as a purifying mechanism to eliminate suboptimal or prematurely differentiating cells[8–13]. In addition, MYC activity, together with that of MYC-N, is essential for maintaining mouse pluripotent cells in the cell cycle and the compound elimination of *Myc* and *Mycn* leads to cell death or diapause[14].

Despite the relevance of MYC function and expression levels in pluripotent cells, there is no information on how *Myc* expression is regulated in this context. Preliminary studies suggested that transcriptional control is the most important regulatory step in determining *Myc* expression levels in mESCs[8]. Indeed, *Myc* transcript and protein show very short half-lives, and changes in *Myc* transcription directly impact on MYC protein levels[10]. The mammalian *Myc* transcription unit is located in a 3 megabases (Mb) gene desert within which, several remote enhancers drive *Myc* transcription in different subtypes of cells, including craniofacial precursors, hematopoietic lineages, and hematological tumors[15–18].

In this study, we use knock-in and knock-out strategies and integrate in vivo and in vitro studies to identify the enhancers that regulate *Myc* transcription in mouse pluripotent cells. We identified a topologically associated region dedicated to *Myc* regulation in pluripotent cells and other stem cells of the early mouse embryo. Within this region, we found specific regulatory elements dedicated to promote *Myc* expression in different cell types and stages of pluripotency.

## Results

### Analysis of *Myc* regulatory regions in early mouse embryos

*Myc* is located within a 3 Mb gene desert in the mouse chromosome 15, closely linked to the *Pvt1* gene, which produces a long non-coding RNA[19,20] (Fig. 1a). To identify genomic regions involved in the regulation of *Myc* expression in pluripotent cells, we determined MYC protein expression in the epiblast of E6.5 mouse embryos with engineered chromosome rearrangements affecting the *Myc* gene desert[16]. These rearrangements delete large segments of the locus or, in the case of *InvMyc1*, removes the regulatory regions downstream of *Pvt1* away from the *Myc* promoter (Fig. 1b–h). We found normal *Myc* expression in all rearrangements analyzed (Fig. 1a–h). Altogether, the regulatory regions tested included most of the 3 Mb gene desert, except distal 5′ regions and a 300 kilobase (kb) region that contained the *Myc* and the

*Pvt1* transcription units. In agreement with this functional analysis, the published epigenetic landscape around *Myc* in mouse ESCs shows that the H3K27 acetylation marks, indicative of active transcription/enhancer activity, are present within a ~250 kb region spanning from ~10 kb upstream to ~240 kb downstream the *Myc* transcriptional start (Fig. 1i; Supplementary Fig. 1a). The distribution of other epigenetic marks (H3K4me1 and P300), of the ES-cell transcriptional regulator FOXD3 and of the core pluripotency factors OCT4, NANOG and SOX2 are also enriched in this ~210 kb region (Fig. 1i; Supplementary Fig. 1a)[21–27]. Furthermore, Hi-C maps of mouse ES cells[28,29] show that *Myc* is located at the junction of two large Topologically Associated Domains (TAD) (Supplementary Fig. 1b), with a sub-TAD that coincides with this ~210 kb region (Supplementary Fig. 1b). We therefore focused our attention on this potential regulatory region.

### BAC transgene-driven *Myc* embryonic expression

We identified a Bacterial Artificial Chromosome (BAC RP24-78D24) that covers the ~250 kb candidate region. This BAC has a total length of 231 Kb including ~28 kb upstream the *Myc* transcriptional start and ~203 kb downstream (Fig. 1a, i). To study whether the BAC DNA contained regulatory regions relevant for *Myc* expression in pluripotent cells, we designed a knock-in strategy to target the *Myc* coding region. Given the fast turnover of *Myc* transcripts and protein, and the dynamic expression of *Myc* in early mouse development[30–32], we reasoned that capturing *Myc* transcriptional regulation would require a reporter with the same dynamics as the endogenous protein. The long half-life of fluorescent proteins makes them unsuitable for these purposes; in contrast, a GFP knock-in that produces a GFP-MYC fusion protein[33] accurately reports *Myc* expression fast dynamics in vivo and in vitro[10,34] (Fig. 2a). Based on this, we designed a similar knock-in strategy to insert the Turquoise Fluorescent Protein (TFP) in frame with the MYC ATG starting codon[35], which is predicted to produce a TFP-MYC fusion protein from the BAC *Myc* transcription unit (Fig. 2a).

We next generated transgenic mice carrying the BAC, derived ESCs from them, and characterized the transgenic insertion by whole-genome sequencing. Copy number variation detection (Supplementary Fig. 2a, b) showed insertion of one complete copy plus a second copy with a 35 kb deletion next to a fragment of chromosome 19 (Supplementary Fig. 2f). Fluorescent In Situ Hybridization (FISH) using the BAC DNA showed a single site insertion at chromosome 15 at an estimated distance of ~38 Mb from the endogenous *Myc* locus (Supplementary Fig. 2c–e). Given that the transgene had been inserted in chromosome 15, where the endogenous locus resides, we performed Circular Chromosome Conformation Capture (4C) analyses using as viewpoints the TFP sequence in *Myc^+/+^*, *Myc^2TFP+^* ES cells, to report the interactions of the transgenic *Myc*, and the GFP sequences in *Myc^GFP/GFP^* cells, to report the interactions of the endogenous locus (Supplementary Fig. 3). Interactions of TFP with the *Myc* regulatory sequences were found up to the limits of the BAC DNA, but not beyond (Supplementary Fig. 3c). In contrast, the GFP viewpoint showed interactions with the *Myc* regulatory regions beyond the limits of the BAC DNA boundaries (Supplementary Fig. 3b). This indicated that the *TFP-Myc* transgene does not have access to endogenous regulatory elements but preserves interactions within the BAC DNA. Within the common sequences between the BAC and the endogenous locus, the interaction profile was similar except for the *Pvt1* promoter, which shows a much stronger interaction in the endogenous locus than in the BAC transgene. This might be related to the fact that the BAC sequences only contain part of the *Pvt1* transcriptional unit (Fig. 1a, i).

We then confirmed the expression of a 95 kDa TFP-MYC fusion protein equivalent to GFP-MYC, whereas the Wild-Type (WT) MYC protein has 65 kDa (Supplementary Fig. 4). Next, we performed a study of *TFP-Myc* expression compared to that of *GFP-Myc* during early mouse embryo development (Fig. 2b–g′). At E3.5, *TFP-Myc* shows an expression pattern similar to that of *GFP-Myc*, being expressed at

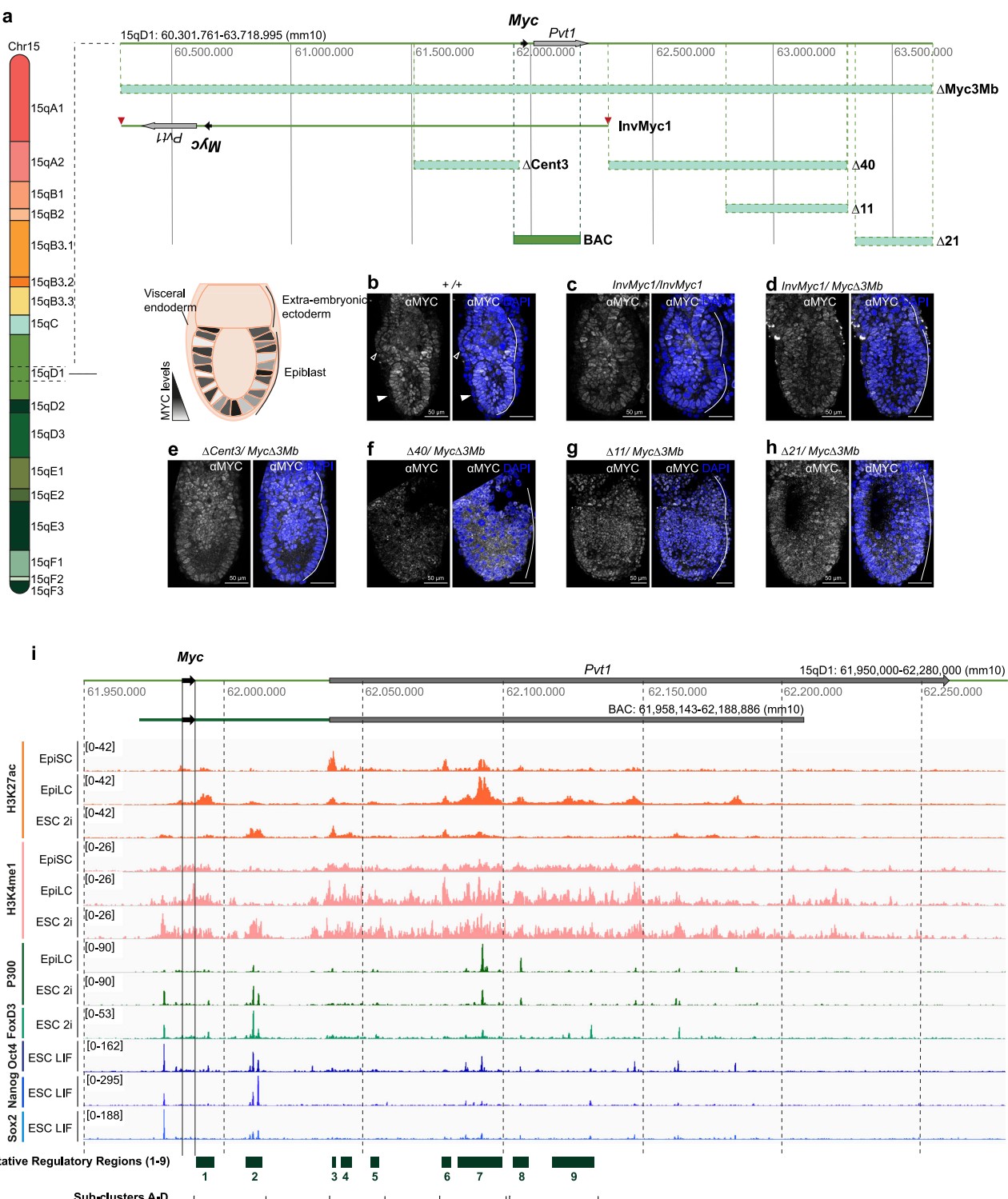

**Fig. 1 | Chromosome rearrangement analysis in the 3 Mb gene desert and epigenetic landscape of the *Myc* locus and downstream sequences.**
**a** Representation of the mouse genomic region containing the *Myc* and *Pvt1* transcription units and its chromosomal location. Chromosome rearrangements and the BAC RP24-78D24 are also represented. **b**–**h** Immunofluorescence against MYC protein in WT (**b**) and genomic rearrangement-containing E6.5 embryos, as indicated. Scale bar = 50 microns. *N* = 9 WT, 17 Inv1, 11 Inv1xMycΔ3 Mb, 13 ΔCent3, 7 Δ40, 8 Δ11, 8 Δ21. Empty arrowheads indicate *Myc* expression in the extraembryonic visceral endoderm; solid arrowheads indicate the absence of *Myc* expression in the embryonic visceral endoderm. **i** Scaled representation of the *Myc* transcriptional unit (black arrow) and *Pvt1* (gray arrow) in the genome and in BAC RP24-78D24. Below, H3K27ac (orange) and H3K4me1 (pink) distribution in EpiSCs (primed pluripotency), EpiLCs (formative pluripotency), and ESCs (naive pluripotency). P300 (dark green) binding in EpiLCs and ESCs (ref.). FOXD3 (light green), OCT4, NANOG and SOX2 (blue) binding in ESCs[21,26]. Putative cis-regulatory regions are shown as green boxes and classified into sub-clusters A to D.

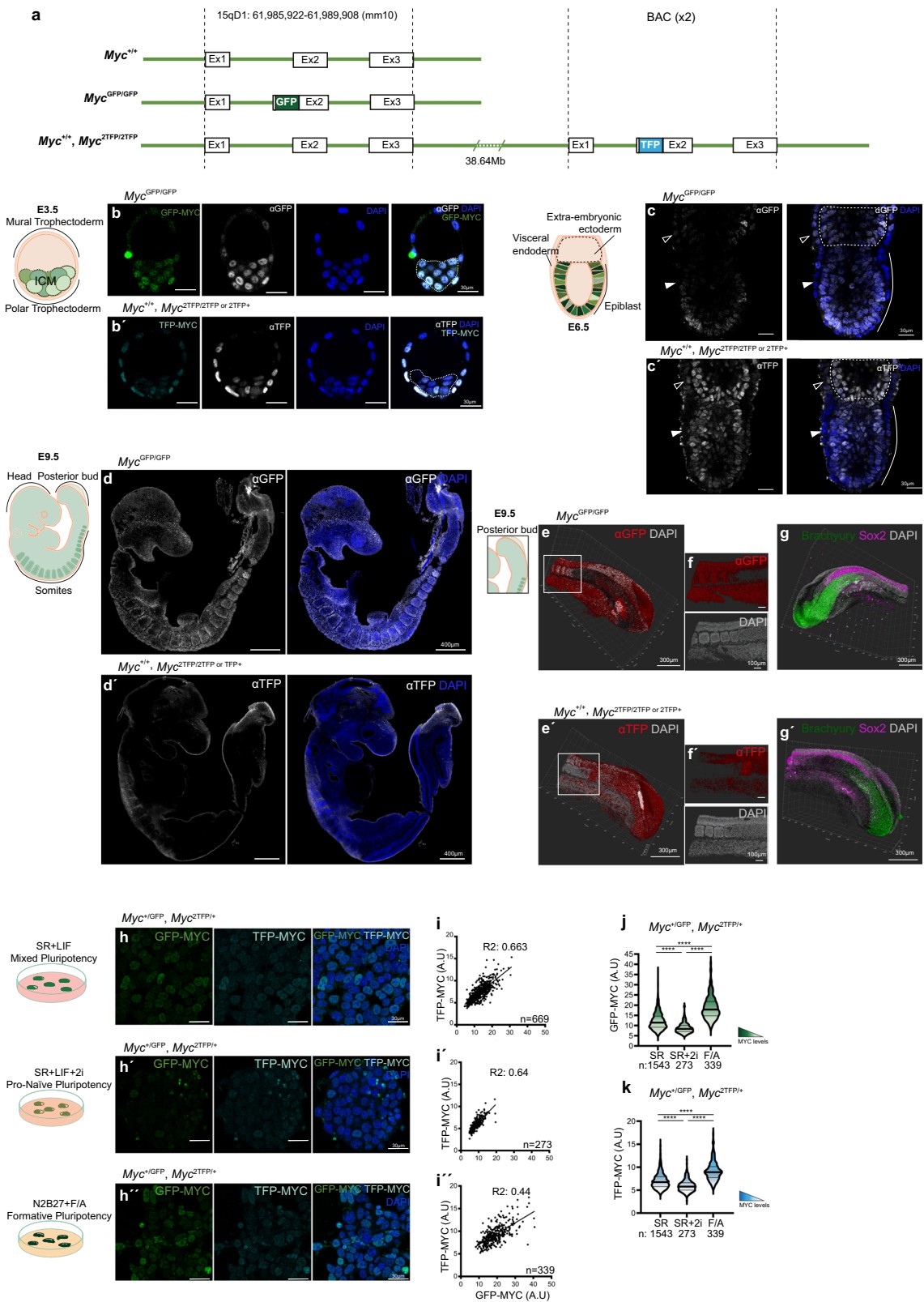

heterogeneous levels in the Inner Cell Mass (ICM) and in the tropho-blast, with higher expression in polar than in mural trophectoderm (Fig. 2b, b´). At E5.5–6.5, *TFP-Myc* shows a similar expression pattern to *GFP-Myc* in the epiblast, in extraembryonic ectoderm, and in the extraembryonic part of the visceral endoderm (Fig. 2c, c´). In contrast, *TFP-Myc* expression strongly diverges from that of *GFP-Myc* in E9.5-

E10.5 embryos (Fig. 2d–g; Supplementary Fig. 5). *GFP-Myc* is widely expressed in embryonic tissues with higher expression in the fore-brain, nasal processes, branchial arches, limb buds and somites and somewhat lower expression levels in the posterior embryonic bud. In contrast, *TFP-Myc* is only expressed in the posterior embryonic bud (Fig. 2d–g; Supplementary Fig. 5a–f). In 3D analysis, both TFP-MYC and

**Fig. 2 | Analysis of the regulatory activity of BAC RP24-78D24 sequences in mouse embryos and ESCs. a** Schemes of the WT *Myc*<sup>+/+</sup>; WT *Myc*<sup>GFP/GFP</sup> and *Myc*<sup>+/+</sup>, *Myc*<sup>2TFP/2TFP</sup> genotypes. Confocal images showing the endogenous *Myc* expression pattern in E3.5 blastocysts revealed by GFP-MYC (**b**) and the transgenic *Myc* expression pattern revealed by TFP-MYC (**b´**) using simultaneous detection of native fluorescence and immunostaining against GFP/TFP. Scale bar = 30 microns. *N* = 5 *Myc*<sup>GFP/GFP</sup> and 13 *Myc*<sup>+/+</sup>; *Myc*<sup>2TFP/2TFP</sup> embryos. **c, c´** Confocal images showing immunostaining against GFP/TFP in GFP-MYC and TFP-MYC E6.5 embryos. Scale bar = 30 microns. *N* = 8 *Myc*<sup>GFP/GFP</sup> and 9 *Myc*<sup>+/+</sup>; *Myc*<sup>2TFP/2TFP</sup> embryos. Empty arrowheads indicate *Myc* expression in the extraembryonic visceral endoderm; solid arrowheads indicate the absence of *Myc* expression in the embryonic visceral endoderm. **d, d´** Confocal images showing immunostainings against GFP/TFP in GFP-MYC and TFP-MYC E9.5 embryos. Scale bar = 400 microns. *N* = 6 *Myc*<sup>GFP/GFP</sup> and 10 *Myc*<sup>+/+</sup>; *Myc*<sup>2TFP/2TFP</sup> embryos. **e–g** 3D reconstruction of the posterior embryonic buds of embryos shown in (**d, d´**). **e, e´** Imaris 3D reconstruction show immunostaining against GFP/TFP of GFP-MYC and TFP-MYC. Scale bar = 300 microns.

**f, f´**, Magnification of E9.5 somitic and pre-somitic region of *Myc*<sup>GFP/GFP</sup> and *Myc*<sup>+/+</sup>; *Myc*<sup>2TFP/2TFP</sup> embryos. Scale bar = 100 microns. **g, g´** 3D reconstructions show immunostaining with anti-Brachyury + anti-SOX2 in the posterior bud. Scale bar = 300 microns. *N* = 3 *Myc*<sup>GFP/GFP</sup> and 3 *Myc*<sup>+/+</sup>; *Myc*<sup>2TFP/2TFP</sup> embryos. **h, h´´**, Confocal images show GFP-MYC and TFP-MYC endogenous expression levels in *Myc*<sup>+/GFP</sup>; s*Myc*<sup>2TFP/+</sup> mESCs cultured in SR + LIF (**h**), in SR + LIF+2i (**h´**) and in N2B27 + F/A (**h´´**). Scale bar = 30 microns. Two independent clones were analyzed. **i, i´´** Plots showing the correlation between GFP-MYC and TFP-MYC signals at the single-cell level in SR + LIF, SR + LIF+2i, and N2B27 + F/A culture conditions. The regression line is represented, and the coefficient of determination (R squared) is shown for each condition. The number of cells analyzed is shown in the graphs. **j, k** Violin plots show GFP-MYC (green) and TFP-MYC (blue) endogenous signals with median and quartiles in SR + LIF, SR + LIF+2i, and N2B27 + F/A culture conditions. Number of cells (*n*) is shown. Mann–Whitney test, two-sided; ns: *P*-value > 0,05, *****P*-value < 0,0001. Source data for all graphs are available from the Source Data file and raw data from Figshare (see "Data availability" section).

GFP-MYC signals are found in a region coincident with the localization of a multipotent stem cell population known as Neuro-Mesodermal Progenitor (NMPs)[36], identified by double staining with SOX2 and Brachyury (T) (Fig. 2e–g; Supplementary Fig. 5g–j). We also observed a strong TFP-MYC signal in the Pre-Somitic Mesoderm (PSM) which contains precursors of somites (Fig. 2f, f´). The precursors of the lateral plate mesoderm and neural tube also show expression that fades rapidly as the tissues are incorporated into the embryonic axis[37,38]. These results show that, besides enhancers active in pluripotent embryonic cells, the BAC sequences include regulatory regions able to activate *Myc* expression in multipotent embryonic and extra-embryonic stem and precursor cell populations. In contrast, the BAC does not drive expression in the more differentiated tissues of the E9.5 embryo. While we cannot exclude that some late enhancers within the BAC are repressed by the genomic insertion environment, these results strongly suggest that enhancers required for *Myc* expression in more differentiated cells are likely located further away, as suggested by previous analysis of the endogenous locus[18].

We next studied the expression of TFP-MYC in mouse ESCs, which allow in vitro induction of different states of pluripotency in transit to differentiation. We studied *Myc*<sup>GFP/GFP</sup> and *Myc*<sup>+/GFP</sup>, *Myc*<sup>+/2TFP</sup> cells in culture conditions that promote either naive pluripotency (SR + LIF+2i)[39,40], the formative pluripotency in transition to primed (2 days in N2B27 + F/A, hereafter N2B27 + F/A)[7] or mixed pluripotency (SR + LIF)[10] (Supplementary Fig. 6a; Fig. 2h–k).

We then analyzed *TFP-Myc* expression in the 3 culture conditions. We observed that the global distribution of *TFP-Myc* expression levels was similar to that observed for *GFP-Myc* (Fig. 2h, j, k). After two days in culture in N2B27 + F/A conditions cells attain the EpiLC state (formative pluripotency) and express the highest TFP-MYC/GFP-MYC levels (Fig. 2h´´, j, k). In contrast, SR + LIF+2i promotes the lowest levels (Fig. 2h´, j, k), while the SR + LIF condition promotes intermediate levels (Fig. 2h, j, k). The presence of the two reporters in the same ESC line allowed us to compare endogenous and transgenic MYC levels at the single-cell level. We found a positive correlation between the two signals that was stronger in SR + LIF+2i and SR + LIF conditions and less so in N2B27 + F/A conditions (Fig. 2i–i´´). We also analyzed *GFP-Myc* or *TFP-Myc* expression levels in *Myc*<sup>GFP/GFP</sup>, *Myc*<sup>+/+</sup>, and *Myc*<sup>2TFP+</sup> ESCs, and we observed similar behaviors between the *TFP* and *GFP* alleles (Supplementary Fig. 6b–e). We conclude that the BAC sequences are sufficient to drive *Myc* expression in the different states of mESC pluripotency and contain sequences required for the fine tuning of *Myc* expression in pluripotent stem cells.

The analysis of the regulatory activity of the sequences in the BAC thus identifies a large region dedicated to regulating *Myc* expression in pluripotent and multipotent stem cells of the early embryo. We therefore named this regulatory region "Early Embryonic Expression" (EEE).

## Modular activity of *Myc* enhancers during pluripotency

We next wanted to identify specific regions involved in the regulation of *Myc* in pluripotent stem cells and their functions. A meta-analysis of published epigenetic data of the *Myc* regulatory regions in pluripotent stem cells identified putative enhancers within the region of interest[22–27]. We focused on regions showing high H3K27Ac and p300 signal in at least one of these three cell types: ESCs, EpiSCs and EpiLC, with preference for those that show a dynamic profile between these cell types, potentially indicating the presence of enhancers that regulate *Myc* expression in the different phases of pluripotency (represented by green boxes in Fig. 1i). We then induced CRISPR-Cas9 deletions in *Myc*<sup>GFP/GFP</sup> ESCs to delete 4 sub-clusters A-D, each containing 2–3 of the putative enhancer-containing regions (Fig. 1i). *GFP-Myc* expression analyses were then performed in the three pluripotency conditions described above. While we obtained homozygous deletions for sub-clusters B and C, we could only retrieve heterozygous deletions for A and D. Only cells with the deletions in sub-cluster C presented lower GFP-MYC level than controls in a statistically significant manner (Fig. 3a–g´´). Expression was reduced in SR + LIF and N2B27 + F/A conditions but not in SR + LIF+2i (Fig. 3). The reduction in expression for sub-cluster C deletion was more intense in the N2B27 + F/A culture condition, which is in accordance with the dynamics of epigenetic marks in the sub-cluster C region, where activating marks are more strongly detected in EpiLCs (Figs. 2i, 3g, h). In addition to this reduction, we observed an increase of *GFP-Myc* expression in sub-cluster B-deleted cells in SR + LIF and N2B27 + F/A conditions (Fig. 3g, g´´), which suggests that instead of enhancers, this region contains elements that limit *Myc* expression levels.

The expression profile of sub-cluster C KO cells in the different culture conditions is, therefore, significantly altered (Fig. 3h). While we could not obtain sub-cluster A or D homozygous deletions, sub-cluster C deletion in heterozygosity provokes detectable MYC level reduction in SR + LIF and N2B27 + F/A conditions (Supplementary Fig. 7a–e), suggesting that if sub-clusters A and D contained enhancer activity similar or stronger than that of sub-cluster C in these culture conditions, it would have been detected. Although we failed to detect a naive-specific enhancer using this approach, we noticed that heterozygous sub-cluster A deletion produced a non-significant tendency to reduce expression levels specifically at the naive- and mixed-pluripotency states, whereas it does not affect expression in N2B27 + F/A condition (Fig. 3g–g´´). In addition, this observation correlates with the increased chromatin activation of enhancer-2 in mESCs cultured in SR + LIF+2i (Fig. 4a, enhancer-2).

We next analyzed putative individual enhancers within sub-cluster C (Fig. 4a–f; Supplementary Fig. 7f–m). We identified 4 putative enhancers of ~3 kb each (7-1 to 7-4 in Fig. 4a), based on the epigenetic marks and the ENCODE scores for Candidate Cis-regulatory Elements

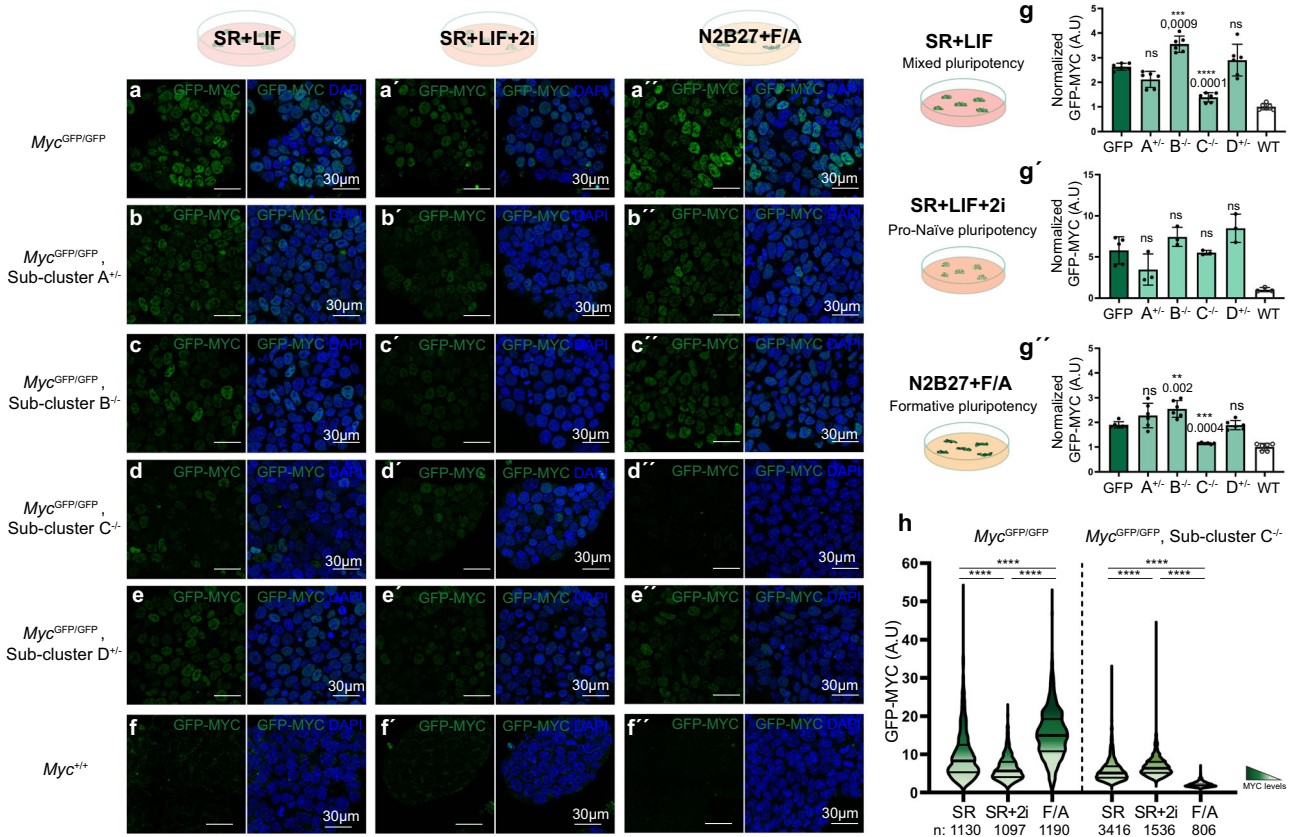

**Fig. 3 | Deletion analysis of endogenous cis-regulatory regions in the BAC RP24-78D24 region.** Confocal images showing GFP-MYC endogenous fluorescence and DAPI in control *Myc^GFP/GFP* cells cultured in SR + LIF (**a**), SR + LIF+2i (**a´**), and N2B27 + F/A (**a´´**). *Myc^GFP/GFP* ES cells with heterozygous deletion of sub-cluster A (**b, b´´**), homozygous deletion of sub-cluster B (**c, c´´**), homozygous deletion of sub-cluster C (**d, d´´**) and heterozygous deletion of Sub-cluster D (**e, e´´**) and WT *Myc^+/+* ES cells (**f, f´´**) cultured in the same 3 conditions. Scale bar = 30 microns. *N* = 3 clones for each genotype. **g, g´´** Graphs showing the normalized median intensity of 3 clones for each genetic condition with 2 biological replicates for each clone. **g´** 3 independent clones for all conditions except for the GFP condition in which 5 independent clones were used. The number of cells quantified per clone/replicate is available from the source data provided. GFP-MYC levels in

SR + LIF and N2B27 + F/A conditions were analyzed by flow cytometry (**g, g´´**) and by confocal imaging in SR + LIF+2i condition (**g´**). Each dot represents the median of an individual clone/biological replicate and bars indicate the mean ± standard deviation of all clones/replicate in each condition. The number of cells quantified per clone/biological replicate is available from the Source Data in Figshare. One-way ANOVA with Dunnett's correction; ns: *P*-value > 0,05; **P*-value < 0,01, ***P*-value < 0,001, ****P*-value < 0,0001. **h** Violin plots with median and quartiles show GFP-MYC intensity in control *Myc^GFP/GFP* and sub-cluster C KO cell populations cultured in three conditions, as indicated. The number of cells analyzed is shown below the graphs. Mann–Whitney test, two-sided; ****P*-value < 0,0001. Source data for all graphs are available from the Source Data file and raw data from Figshare (see "Data availability" section).

(cCREs) (Supplementary Fig. 7f). We also incorporated in this analysis a 6 kb deletion of enhancer-2 from the sub-cluster A region. We performed CRISPR-Cas9 homozygous deletions of each enhancer in GFP-MYC cells and characterized their expression in the three established conditions. We only detected altered expression in cells with enhancer-2 and enhancer-7-3 deletions (Fig. 4b–f; Supplementary Fig. 7g–m). Enhancer-2 KO mESCs presented reduced GFP-MYC levels in SR + LIF+2i, a milder reduction of expression in SR + LIF and non-significant expression changes in N2B27-F/A conditions (Fig. 4b, c´´, e, f). Enhancer-7-3 KOs showed a complementary behavior; strongly reduced GFP-MYC levels in N2B27 + F/A, a milder reduction of expression in SR + LIF conditions and non-significant expression changes in SR + LIF+2i conditions (Fig. 4b´´, d, d´´, e, f). These results suggest that the 6 kb enhancer-2 is preferentially dedicated to regulating *Myc* expression in cells at the naive pluripotency state, while the 3 kb enhancer-7-3 preferentially regulates *Myc* expression in cells at an EpiLC/formative pluripotency state.

To determine whether the in vitro results correlate with enhancer expression dynamics in vivo, we deleted enhancer-2 and enhancer-7-3 in *Myc^GFP/GFP* knock-in mice, using the same CRISPR-Cas9 deletion strategy as in mESCs (see "Methods" section and Fig. 5). We then

generated embryos carrying the deletions in homozygosity and analyzed them in comparison with *Myc^GFP/GFP* homozygous embryos with intact regulatory sequences (Fig. 5a). Preimplantation embryos (E3.5) with enhancer-2 deleted show reduced *GFP-Myc* expression in epiblast cells of the blastocyst (Fig. 5b, c, e; Supplementary Fig. 8a), whereas embryos with enhancer-7-3 deleted showed normal *GFP-Myc* expression in the blastocyst (Fig. 5b, d, f; Supplementary Fig. 8b). The reduction of *GFP-Myc* expression in enhancer-2 deleted embryos is specific to epiblast, not affecting the trophectoderm (Fig. 5b, c, e; Supplementary Fig. 8a). In contrast, embryos homozygous for the enhancer-2 deletion did not show any alteration of GFP-MYC levels in post-implantation embryos (Fig. 5g–j´; m; Supplementary Fig. 8c), whereas post-implantation embryos homozygous for the enhancer-7-3 deletion showed a strong and specific reduction of GFP-MYC levels in the epiblast (Fig. 5g, h´; k, l´; n Supplementary Fig. 8d, e–g). This reduction is only present in epiblast cells and not in other regions of the embryo in which *Myc* is strongly expressed, like the extra-embryonic ectoderm and extraembryonic visceral endoderm (Fig. 5g, h´; k, l´; n; Supplementary Fig. 8d, e–g). Further analysis of *GFP-Myc* expression in enhancer-7-3 KO embryos at E9.5 showed no differences with control embryos (Supplementary Fig. 8h–s). These results

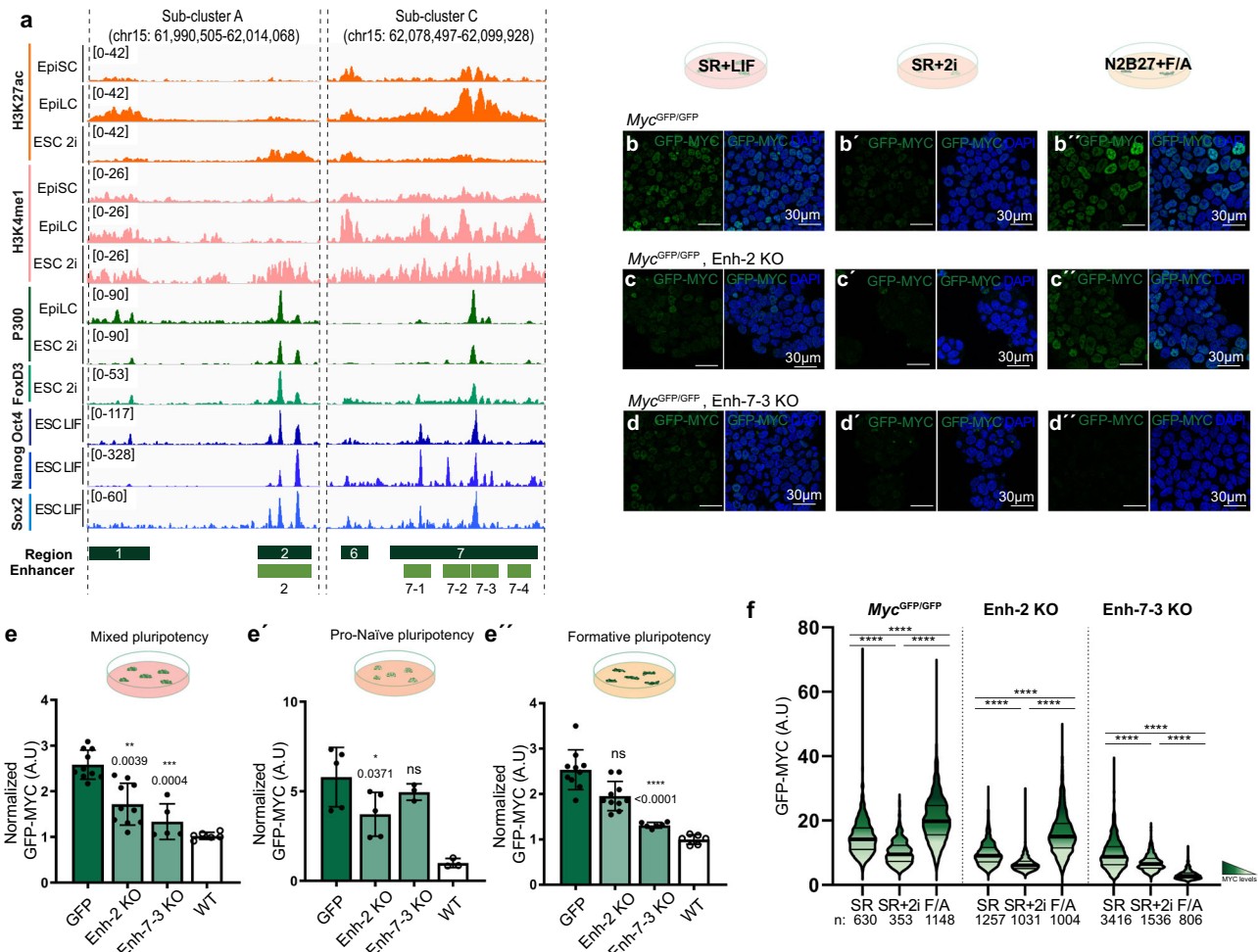

**Fig. 4 | Identification of enhancers that regulate *Myc* in different states of pluripotency. a** Epigenetic landscapes of sub-clusters A and C in EpiSCs, EpiLCs, and ESCs-2i, as previously described. H3K27ac (orange); H3K4me1 (pink); P300 (dark green); FOXD3 (light green); OCT4-NANOG-SOX2 (blue). Data from the same sources as in Fig. 1. Putative cis-regulatory regions are represented as dark green boxes and numbered according to Fig. 1. Subdivisions of these regions into 5 putative smaller regulatory regions are represented as light green boxes: 2 (Sub-cluster A); 7-1; 7-2; 7-3; 7-4 (Sub-cluster C). Confocal images show GFP-MYC endogenous fluorescence in control *Myc*<sup>GFP/GFP</sup> cells (**b, b´´**) and cells deleted for enhancer-2 (Enh2<sup>−/−</sup>) (**c, c´´**) and enhancer-7-3 (Enh-7-3<sup>−/−</sup>) (**d, d´´**) in SR + LIF (**b, c, d**), SR + LIF+2i (**b´, c´, d´**) and N2B27 + F/A (**b´´, c´´, d´´**). Scale bar = 30 microns. *N* = 3 clones per genotype. **e, e´´**, Dot plot with bar shows normalized median intensity of *Myc*<sup>GFP/GFP</sup> clones. GFP-MYC endogenous levels were analyzed by FACs in SR (**e**) and N2B27 + F/A (**e´´**). In SR + LIF+2i condition (**e´**) GFP-MYC levels

were analyzed in confocal images. Each dot represents the median of an individual clone/replicate and bars indicate the mean ± standard deviation of all clones/replicate in each condition. The number of cells quantified per clone/biological replicate is available from the source data in Figshare. **e, e´´**, independent clones per condition: 5 GFP, 5 Enh-2 KO, 3 Enh-2 KO, 3 WT each with two biological replicates. **e´** independent clones per condition: 5 GFP, 5 Enh-2 KO, 3 Enh-2 KO, 3 WT. One-way ANOVA with Dunnett's correction; ns: *P*-value > 0,05; *P*-value < 0,05; ***P*-value < 0,01; ****P*-value < 0,0001. **f** Violin plots with median and quartiles show GFP-MYC endogenous signals in control *Myc*<sup>GFP/GFP</sup> cells and in cells with homozygous deletions of enhancer-2 or enhancer-7-3 cultured in the three culture conditions, as indicated. The number of cells analyzed is shown below the graphs. Mann–Whitney test, two-sided; ****P*-value < 0,0001. Source data for all graphs are available from the Source Data file and raw data from Figshare (see "Data availability" section).

confirmed the specificity of enhancers-2 and 7-3 in regulating *Myc* expression in pluripotent cells in vivo. Furthermore, each enhancer specifically affects *Myc* expression at a stage of epiblast development in vivo that correlates with their specificity for *Myc* expression in cultured cells under different culture conditions. These analyses show that enhancer-7-3 is specifically dedicated to a late phase of pluripotency characteristic of the post-implantation epiblast, however it does not contain sequences required for *Myc* expression in earlier phases of pluripotency or in other lineages of early mouse embryos. Conversely, enhancer-2 is active during naive pluripotency but not in pluripotent cells of the post-implantation embryo. Based on these results we renamed enhancer-2 as NPE, for Naïve Pluripotency Enhancer, and enhancer-7-3 as FPE, for Formative Pluripotency Enhancer.

**Functional dissection of NPE and FPE *Myc* enhancers**

To study the molecular basis of the different activities of NPE and FPE, we performed a meta-analysis using data from the 3D genome browser (http://3dgenome.fsm.northwestern.edu/) and the ChIP Atlas Database[41,42] (https://chip-atlas.org/peak) (Fig. 6a–c). Proximity ligation-assisted ChIPseq (PLAC-seq)[43] showed dense interactions between the *Myc* promoter and H3K4me3-, H3K27Ac- and Pol2-bound regions in the BAC region 3' to the *Myc* locus and much more sparse interactions beyond this region (Fig. 6a). A closer analysis of the region containing putative pluripotency enhancers shows coincident interactions of H3K27Ac+ and Pol2+ regions in several of these elements, including NPE and FPE (Fig. 6b). The core pluripotency factors: NANOG, SOX2, and OCT4 bind abundantly to two regions of NPE and one region in FPE (Fig. 6b, c), which correlates with their role as core

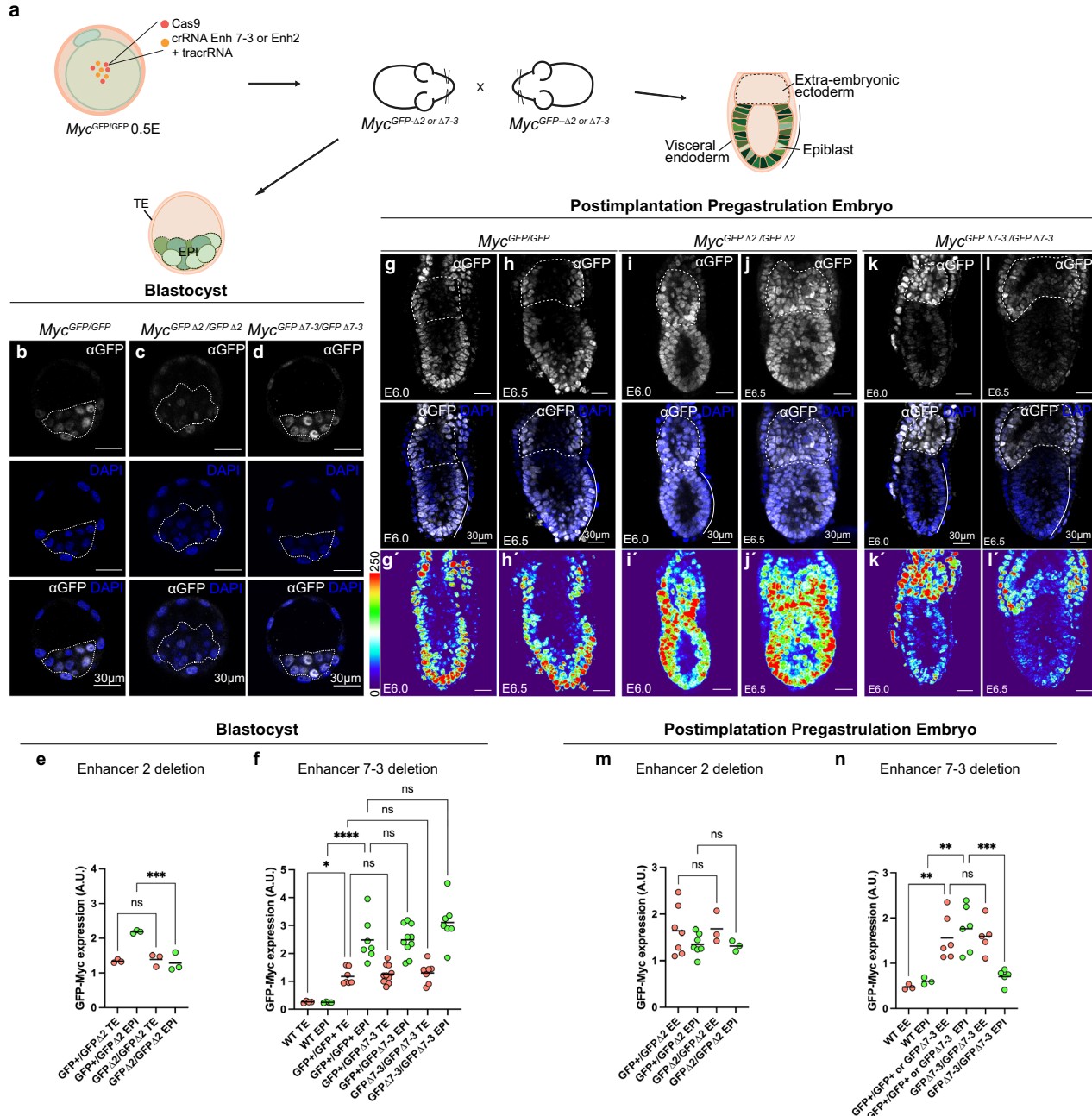

**Fig. 5 | Regulatory activity of naive versus formative pluripotency enhancers in vivo. a** Representation of enhancer-2 and enhancer-7-3 deletions in *Myc*^GFP/GFP fertilized eggs by CRISPR-Cas9. Crosses were then set up to generate *Myc*^GFP/GFP with enhancer-2 (Δ2) or enhancer-7-3 (Δ7-3) deletions in the littermate embryos. Confocal images show immunostaining against GFP and DAPI in E3.5 embryos of genotypes *Myc*^GFP/GFP (**b**), KO for enhancer-2 (**c**), and KO for enhancer-7-3 (**d**). Scale bar = 30 microns. *N* = 8 *Myc*^GFP/GFP, 3 *Myc*^GFP/GFP Δ2/+, 3 *Myc*^GFP/GFP Δ2/Δ2, 10 *Myc*^GFP/GFP Δ7-3/+; 7 *Myc*^GFP/GFP Δ7-3/Δ7-3. Dot plots showing GFP-MYC expression levels in trophectoderm (TE) and epiblast cells (EPI) of *Myc*^GFP/GFP Δ2-heterozygous and Δ2-homozygous embryos (**e**) and in WT, *Myc*^GFP/GFP and *Myc*^GFP/GFP Δ7-3-heterozygous and Δ7-3-homozygous embryos (**f**). Each dot represents the mean intensity per cell measured in individual embryos. The number of cells analyzed by embryo/tissue is provided in Supplementary Fig. 8. For **e** and **f**, ordinary one-way ANOVA with Šídák's multiple comparisons test and two-tailed *P*-values; ns: *P*-value > 0,05;

*P-value = 0.0363; ***P*-value = 0.0003; ****P*-value < 0,0001. Confocal images show immunostaining against GFP and DAPI in post-implantation embryos of control *Myc*^GFP/GFP (**g**), complete KO for enhancer-2 (Δ2) (**i, j**) or for enhancer-7-3 (Δ7-3) (**k, l**). Scale bar = 30 microns. **g´–l´**, Heatmap representation of the anti-GFP channel of each embryo shown in **g–l**. Scale bar = 30 microns. *N* = 3 *Myc*^GFP/GFP, 7 *Myc*^GFP/GFP Δ2/+, 3 *Myc*^GFP/GFP Δ2/Δ2, 6 *Myc*^GFP/GFP Δ7-3/+ and 5 *Myc*^GFP/GFP Δ7-3/Δ7-3 embryos. Dot plot shows GFP-MYC expression levels in extraembryonic ectoderm (EE) and epiblast cells (EPI) in Δ2 (**m**) and in *WT*, *Myc*^GFP/GFP, and Δ7-3 (**n**) embryos. Each dot represents the mean intensity per cell measured in individual embryos. The number of cells analyzed by embryo/tissue is provided in Supplementary Fig. 8. Ordinary one-way ANOVA with Šídák's multiple comparisons test; ns: *P*-value > 0,05; **P*-value = 0.0024 for EE and 0.0012 for EPI; ***P*-value < 0.0006. Source data for all graphs are available from the Source Data file and raw data from Figshare (see "Data availability" section).

pluripotency factors responsible for maintaining all pluripotency states from pre-implantation to gastrulation[44–50]. Interestingly, we also found in both enhancers the transcriptional/epigenetic regulator BRD4, required for the specification of the epiblast lineage[51].

In addition, we found factors differentially bound to each enhancer (Fig. 6b, c). NPE is bound in ESCs by ESRRB, FOXD3, KLF4/5 and PRDM14. These factors bind naive pluripotency-specific enhancers in ESCs[22], while ESRRB and KLF4/5 regulate specific characteristics of

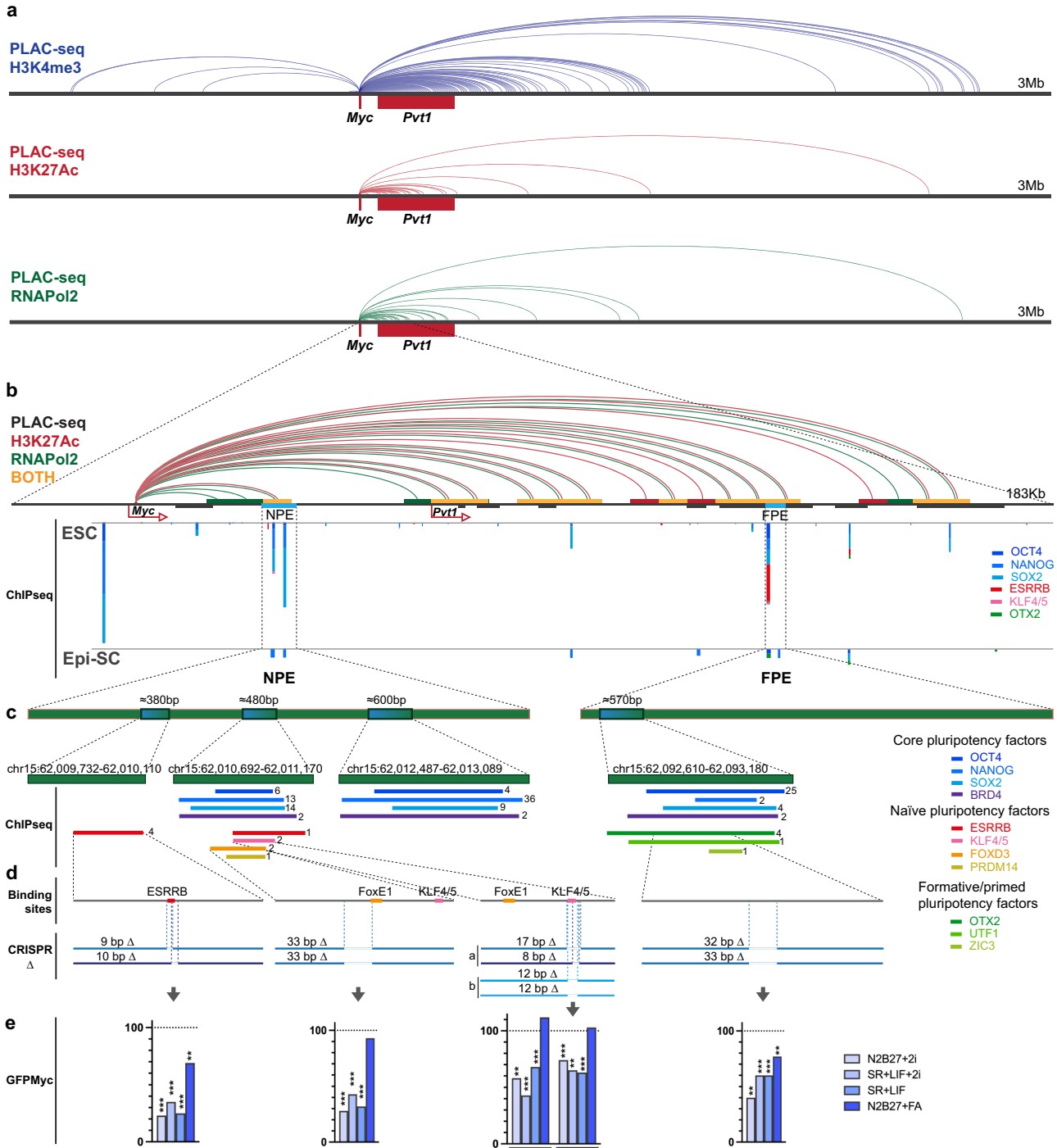

**Fig. 6 | Transcription factors responsible for the regulatory activity of NPE and FPE. a** PLAC-seq data showing the interaction of the *Myc* promoter with H3K4me3-, H3K27Ac- and Pol2-positive regions in the whole *Myc* regulatory region (data from ref. 43 downloaded from the 3D genome browser[68] http://3dgenome. fsm.northwestern.edu/). **b** PLAC-seq data showing the interaction of the *Myc* promoter with H3K27Ac- and Pol2-positive regions and ChIPseq data for pluripotency factors in the regulatory region containing the putative NPE and FPE. Above the line that represents the genome, green color segments indicate H3K4me3 interactions with the *Myc* promoter, red color segments indicate interactions with H3K27-positive regions, and yellow color with both. Below, the ChIPseq meta-analysis is shown for ESCs and EpiSCs. ChIPseq data are from published studies (data retrieved from the ChIP Atlas web browser[42] https://chip-atlas.org/peak_browser).

The height of the bars is proportional to the number of independent studies that detect binding for each factor. **c** Enlargement of the regions in NPE and FPE that bind Pluripotency TFs. The numbers to the right of the bars indicate the number of independent studies that detected the interaction. **d** mapping of pluripotency TF DNA-binding motifs coinciding with the ChIPseq interactions and indication of the CRISPR deletions affecting these motifs. **e** Quantification of the percentage of activity that *Myc*^GFP/GFP ESCs carrying TF binding motif deletions show in comparison with control *Myc*^GFP/GFP ESCs in different culture conditions. See Fig. S10 for details on the biological replicates, number of cells analyzed, and exact *P*-values. Ordinary one-way ANOVA with Šídák's multiple comparisons test; ns: *P*-value > 0,05; **P*-value < 0,01; ***P*-value < 0,001. Source data for all graphs are available from the Source Data file and raw data from Figshare (see "Data availability" section).

naive pluripotency and promote this state[22,24,52–54]. FOXD3 is a repressor that controls the progression from naive to primed states, it binds exclusively to naive active enhancers to promote exit from naive state[26]. FPE specifically binds OTX2, ZIC3, and UTF1 in EpiSCs. OTX2 is a pioneer factor that cooperates with OCT4 at the transition from naive to formative/primed states, which is critical for the establishment of the epiblast lineage. OTX2 is upregulated in early post-implantation embryos until gastrulation in vivo and in formative stem cells in vitro and is necessary for stable expansion of EpiLCs[5,6,22]. ZIC3 binds to cell-type-specific enhancers in primed PSCs and plays a critical role in activating pluripotency TF during ESC differentiation to EpiLC[22,24,55]. UTF1 is a recently described regulatory factor that is involved in ESC differentiation[56,57].

These observations correlate with the differential activity of the two enhancers and identify potential specific factors responsible for their activation. To study the potential roles of the candidate transcription factors, we concentrated on those specifically related to either naive or formative pluripotency. We use the JASPAR database[58] to detect binding motifs for KLF4/5, ESRRB, and FOXD3 coincident with the ChIPseq peaks identified in NPE (Fig. 6d). The OCT4 JASPAR binding motif has a very relaxed definition, and we did not find any occurrence in FPE or NPE. For the functional analysis of the sites identified in NPE, we used CRISPR-Cas9 deletion to eliminate each potential binding site in *GFP-Myc* ESCs (Fig. 6d). To discard off-target mutations, we amplified and sequenced the 3 top off-target sites predicted by CRISPOR (crispor.tefor.net/) for each crRNA and found no DNA sequence alterations (Supplementary Fig. 9). Mutations that affect the ESRRB, KLF4/5 or FOXD3 binding sites impaired *GFP-Myc* expression in N2B27 + 2i, SR + LIF+2i and SR + LIF conditions but none of them eliminated the enhancer activity (Fig. 6e and Supplementary Fig. 10). In contrast, none of the mutations affected *GFP-Myc* expression in N2B27 + F/A conditions, with the exception of a milder reduction in N2B27 + F/A conditions by the ESRRB deletion (Fig. 6e and Supplementary Fig. 10). These results suggest that a combination of naive pluripotency-specific transcription factors collaborate to specifically activate the NPE.

Given that we could not identify a candidate DNA-binding sequence for OTX2 in FPE, we studied a 32 bp deletion affecting the center of the OTX2 ChIPseq peaks (Fig. 6d). While this deletion affected *Myc* expression in N2B27 + F/A conditions, it also reduced *Myc* expression, and to a larger extent, in the other culture conditions (Fig. 6e and Supplementary Fig. 10). This suggests that regulation of FPE is complex, and analysis of yet unidentified transcriptions factors is required.

Finally, we studied the conservation of *Myc* enhancers characterized here and found a remarkably low conservation among vertebrates compared to other *Myc* enhancers previously described[16,18,59,60] (Supplementary Fig. 11). We compared the sequence conservation of the enhancers included in the EEE and those previously described among vertebrate species using PhastCons scores. The EEE enhancers constitute the least conserved group, with lower conservation than most other described enhancers and even below the conservation of *Myc* intronic sequences (Supplementary Fig. 11c). This observation suggests a recent appearance of these enhancers in parallel to the evolution of pluripotency network wiring in eutherians[59–61].

### *Myc* pluripotency enhancers regulate mESCs competitiveness
Given that heterogeneous MYC levels drive cell competition in mouse pluripotent cells[8–13], we tested the ability of *Myc* enhancer-deleted mESCs to outcompete their wild-type neighbors. To assay for the competitive ability of the mutant cells, we co-cultured *Myc*^GFP/GFP*;ROSA26R^TdTomato* cells with *Myc*^GFP/GFP* WT cells or *Myc*^GFP/GFP* cell clones deleted for sub-cluster B, which does not increase MYC levels in mESCs, or sub-cluster C, which affects *Myc* expression levels in mixed-pluripotency and EpiLC conditions. *Myc*^GFP/GFP*;ROSA26R^TdTomato* is

a *Myc*^GFP/GFP* WT mESC clone with a TdTomato reporter permanently expressed in the plasma membrane[10]. The Tomato reporter allowed us to easily differentiate the two cell populations in co-culture (Fig. 7a). The co-cultures were maintained during 5 days under SR + LIF or N2B27 + F/A culture media, and the proportion of *TdTomato*-negative cells was quantified every day by flow cytometry (Fig. 7b). The competitive ability of the various clones tested is then deduced from their ability to expand in confrontation with the same *Myc*^GFP/GFP*;ROSA26R^TdTomato* control cells.

We first determined the profile of *GFP-Myc* expression in the control and test cell populations during 5 days in culture in SR + LIF or N2B27 + F/A. We found that GFP-MYC expression levels remained stable in SR + LIF conditions, while they showed widely dynamic changes during the 5 days of culture in N2B27 + F/A (Fig. 7c–e). MYC levels in N2B27 + F/A peaked at day 2, in coincidence with the EpiLC state, and showed a strong downregulation from day 3, as cells differentiate into EpiSCs[7] (Fig. 7f–h). These results match previous studies showing *Myc* downregulation in primed ESCs[10]. GFP-MYC expression levels were similar in the *Myc*^GFP/GFP*;ROSA26R^TdTomato*, the wild-type *Myc*^GFP/GFP*, and the sub-cluster B-deleted *Myc*^GFP/GFP* cells during most of the culturing period and in both culture conditions (Fig. 7c, d, f, g). As expected, cells with homozygous deletions of sub-cluster C showed reduced GFP-MYC levels. In the SR + LIF conditions, the sub-cluster C KO cells presented a reduction of 33.8% on average and this reduction in GFP-MYC levels is similar during the 5 days of culture (Fig. 7e). In the N2B27 + F/A culture conditions, GFP-MYC expression in the sub-cluster C-deleted cells in comparison with *Myc*^GFP/GFP*;ROSA26R^TdTomato* cells showed ~37.2% reduction on average and reached a maximum of ~59% reduction (Fig. 7h).

We then studied the evolution of the proportions of the different cell populations confronted with *Myc*^GFP/GFP*;ROSA26R^TdTomato* cells. The proportion of WT *Myc*^GFP/GFP* cells or *Myc*^GFP/GFP* cells deleted for sub-cluster B was stable during 5 days of co-culture in either SR + LIF or N2B27 + F/A conditions (Fig. 7i, j). In contrast, sub-cluster C-deleted mESCs co-cultured with *Myc*^GFP/GFP*;ROSA26R^TdTomato* cells progressively decreased their proportion during the 5 days of co-culture in either SR + LIF or N2B27 + F/A conditions (Fig. 7i, j). The disadvantage of the sub-cluster C KO cells is greater in the N2B27 + F/A conditions, especially from day 2, in correlation with the strongest imbalance (59% reduction) in GFP-MYC levels between the cell populations (Fig. 7h). In contrast, the proportion of sub-cluster C-deleted ESCs remained stable when populations were cultured independently in either SR + LIF or N2B27 + F/A conditions (Supplementary Fig. 12a, b). These results indicate that the deletion of sub-cluster C renders cells less competitive when confronted with WT cells, in agreement with their lower MYC levels. In addition, increased cell death was observed under both culture conditions in sub-cluster C-deleted ESCs when confronted with WT cells, but not when cultured in isolation (Supplementary Fig. 12c–f), which shows that the lower competitiveness of the mutant cells is, at least in part, due to non-autonomous cell death. These results show that sub-cluster C contains *Myc* regulatory regions essential for mESC competition ability in differentiating conditions.

## Discussion
Here we identified EEE, a regulatory region within a ~200 Kb non-coding DNA downstream of the *Myc* gene that contains cis-acting modules that control *Myc* transcription in mouse pluripotent stem cells and other stem and progenitor cell populations of early mouse embryos. Besides pluripotent cells, the EEE regulates *Myc* expression in the extraembryonic ectoderm, extraembryonic endoderm, and cell populations in the embryonic posterior bud, like NMPs and neural tube- and mesoderm-specific precursors. Within the EEE, we could identify discrete regulatory regions dedicated to pluripotency. Elimination of these regulatory sequences affects *Myc* expression in pluripotent cells but not in precursor/stem cells of the extraembryonic

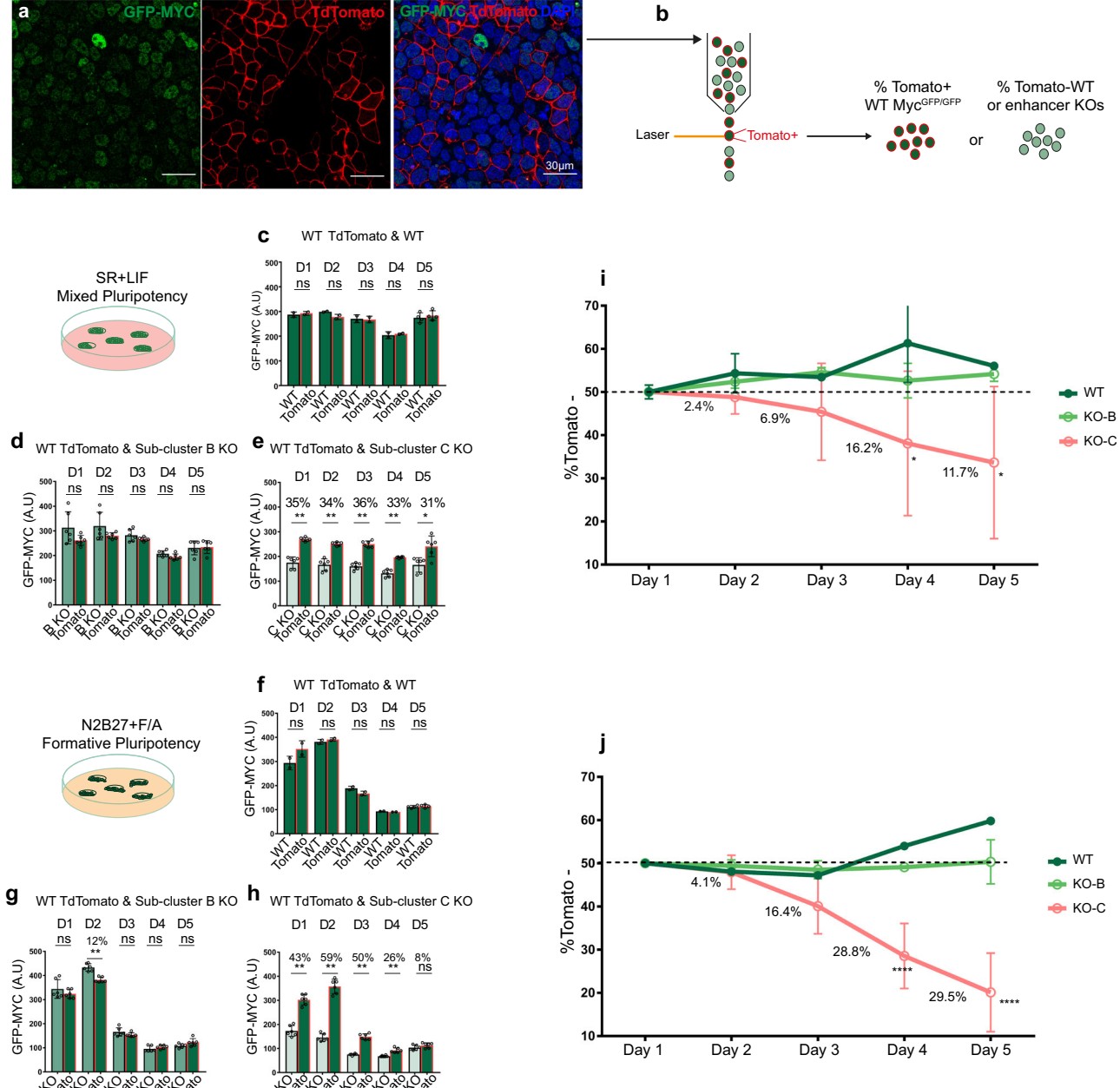

**Fig. 7 | Competitive ability of ESCs defective for *Myc* cis-regulatory elements.**
**a** Confocal images show co-cultures of *Myc^{GFP/GFP}* and *Myc^{GFP/GFP};ROSA26R^{TdTomato}* ES cells. Scale bar = 30 microns. **b** Scheme of the experimental procedure showing cells detached from the co-culture and analyzed by flow cytometry separating Tomato-positive (+) and -negative (−) populations during 5 consecutive days, with measurement of population abundance and GFP-MYC levels. Dot plots with bar show the mean intensity of GFP-MYC endogenous signals of the clones co-cultured in SR + LIF (**c**–**e**) or N2B27 + F/A (**f**–**h**) conditions and analyzed daily by flow cytometry for 5 days. **e, f** *Myc^{GFP/GFP};ROSA26R^{TdTomato}* and co-cultured *Myc^{GFP/GFP}* ESCs, **d, g** *Myc^{GFP/GFP};ROSA26R^{TdTomato}* ESCs and co-cultured *Myc^{GFP/GFP}* ESCs with homozygous deletion of the sub-cluster B (**e, h**), *Myc^{GFP/GFP};ROSA26R^{TdTomato}* ESCs and co-cultured *Myc^{GFP/GFP}* ESCs with homozygous deletion of the sub-cluster C. 3 independent clones with 2 biological replicates each were used for the analyses in (**d, e, g, h**) and two independent clones with biological replicates only on Day 5 in (**c, f**). The number of cells quantified per clone/biological replicate is available from

the source data in Figshare. Mann–Whitney test, two-sided; ns: *P*-value > 0,05; *P*-values for significant differences: **e** Day 1-Day 4 = 0.0022; **g** Day 2 = 0.0022; **h** Day 1-Day 4 = 0.0022. The difference of GFP-MYC levels is shown as percentages significantly different comparisons. Evolution of the tomato- populations is represented as percentage of the whole culture in SR + LIF (**i**) or N2B27 + F/A (**j**) conditions for 5 days. In **i, j**, dots represent mean values and the error bars show the Standard Deviation. 2 different clones were used for WT and cluster B KO and 3 different clones were used for the cluster C KO. The number of cells quantified per clone/biological replicate is available from the source data in Figshare. Day 1 is normalized to 50% and the decrease of the sub-cluster C KO cells is represented as percentage each 24 h. 2-way ANOVA Dunnett's multiple comparisons test; *\*P*-value = 0,0112 for Day 4 and 0.0144 for Day 5; \*\*\*\**P*-value < 0,0001. Non-significant comparisons are not indicated. Source data for all graphs are available from the Source Data file and raw data from Figshare (see "Data availability" section).

regions or posterior embryonic bud. The neighboring non-coding gene *Pvt1* is involved in promoting *Myc* expression in tumor progression[19,62]; however, the BAC used here does not contain the whole *Pvt1* transcript and, while the sub-cluster C and D deletions affect internal *Pvt1* sequences without affecting *Myc* expression, the sub-cluster B deletion eliminates the *Pvt1* promoter and increases *Myc* expression in mixed-pluripotency and formative pluripotency conditions. These results suggest that neither the long-non-coding RNA

encoded by *Pvt1*, nor its transcriptional activity contributes to enhance *Myc* expression in ES cells. Nonetheless, the sub-cluster B contains regions that inhibit *Myc* expression. Interestingly, competition for transcription between the *Myc* and *Pvt1* promoters has been previously described[19] and this phenomenon may underlie the effect of sub-cluster B deletion in pluripotent cells.

One unexpected result was the inability to obtain sub-cluster A and D deletions. This was not due to low-frequency deletion, given that deleted clones in heterozygosity were obtained at normal rates, which suggests lethality or impairment caused by the heterozygous deletions. This phenomenon, however, cannot be attributed to *Myc* loss of function, given that homozygous *Myc* deletion does not compromise ESC renewal or viability[14,63,64] and therefore should involve some unknown mechanism.

Our observations indicate a modular organization of the EEE cluster, with specific enhancer regions dedicated to different cell types (Supplementary Fig. 13). Within the regulatory sequences specifically involved in *Myc* regulation in pluripotent cells, we further identified enhancers largely dedicated to regulate *Myc* transcription in either naive or formative pluripotency states. These results indicate that enhancers in this regulatory region mostly show a modular organization with little functional overlap, revealing enhancer specialization dedicated to different phases of pluripotency. While we cannot exclude the participation of other enhancers with minor roles, or acting redundantly, the intensity of the reduction of *Myc* expression in FPE deletion in the formative epiblast and NPE deletion in the naive epiblast suggests that they largely mediate *Myc* transcriptional activity in pluripotent stem cells.

Interestingly, EEE directs *Myc* expression in cells that belong to the earliest pre-implantation and post-implantation mouse embryo, suggesting that the EEE sub-TAD represents a genomic niche permissive for establishing regulatory interactions with the *Myc* promoter in early embryonic stem cells. This organization is reminiscent of BENC[18], which also comprises a series of specific enhancers whose collective activity regulates *Myc* throughout the hematopoietic cellular hierarchy. While other systems rely on a more distributed organization (e.g., *HoxD* limb archipelagos[65]), *Myc* regulation thus exploits compartmentalization by TAD subdivision in regions dedicated to highly related developmental or differentiation processes. This super-modular organization may allow to rapidly and robustly switch between enhancers in settings in which rapid lineage diversification events take place, like early mammalian development or hematopoiesis. This super-modular organization may also determine the large size of the *Myc* cis-regulatory region (Supplementary Fig. 13).

We showed that independent sets of enhancers respond to the signaling context and this relates to the transcription factors engaged by these signals. The core pluripotency network – OCT4, SOX2, and NANOG – are expressed in naive, formative, and primed state cells; therefore, their activity does not explain the different activities of NPE and FPE. Additional stage-specific factors are therefore likely involved in their specificity and here we show that deletion of ESRRB, FOXD3, or KLF5 putative binding sites in NPE preferentially affects *Myc* expression in naive ESCs. While OTX2, ZIC3, and UTF1 bind FPE in EPISCs, the absence of recognizable DNA-binding motifs for these factors hinders the specific characterization of their role in regulating FPE in formative/primed ESCs.

The temporal activity of the *Myc* enhancers provides a direct comparison as well between the stages of pluripotency in vivo and in vitro, which suggests that the formative pluripotency/EpiLC state spans from early implantation until advanced gastrulation. In primed cells, MYC levels dropped equally in WT cells and cells carrying the different deletions characterized here (day 5 in Fig. 7f–h). The rapid shutdown of *Myc* transcript and protein expression in the late

gastrulating epiblast has been described before[8]. It is unknown whether this shutdown involves active repression or just disengagement of the transcriptional activation machinery, given the short half-life of the MYC protein. Our results indicate that either repressive elements are not required for shutting down *Myc* expression upon differentiation or that, if repressive elements are needed, they do not reside in the regions deleted in this study.

The absence of activity of the identified enhancers beyond the early embryonic stages strongly suggests their specific dedication to pluripotent and early embryonic multipotent stem cells. Given the prominent role of *Myc* in tumor formation and that tumor development involves the establishment of tumor stem cells, by de-differentiation or transformation of pre-existing stem cells, it will be very interesting to determine in the future the putative activation and functional involvement of the identified enhancers during tumor formation.

# Methods

## Ethics statement

Animals were handled in accordance with the CNIC Ethics Committee, Spanish laws, and the EU Directive 2010/63/EU for the use of animals in research. All mouse experiments were approved by the Centro Nacional de Investigaciones Cardiovasculares and Universidad Autónoma de Madrid Committees for "Ética y Bienestar Animal" and the area of "Protección Animal" of the Community of Madrid with references PROEX 220/15 and PROEX 144.1/21.

## Generation of FRT-KanR-FRT-TFP plasmid and TFP-Myc BAC recombineering
TFP was isolated from the plasmid mTurquoise2-C1 (Addgene, Plasmid #54648) by digestion with NheI (NEB, #R3131S) and BamHI (NEB, #R0136S). The host vector with "FRT-KanR-FRT" was digested with SpeI (NEB, #R3133S) and NotI (NEB, #R3189S). Ligation overnight (o/n) (T4 DNA ligase, NEB, #M0202L) was used to directionally link the SpeI and NheI free ends of the insert TFP. Then the free ends of the linear product were blunted (Quick Blunting™ Kit, NEB, # E1201S) and ligated for an additional 5 h. The FRT-KanR-FRT-TFP plasmid obtained was later used to amplify the cassette of interest by PCR for BAC recombineering.

To integrate TFP in frame with the *Myc* locus, we first electroporated the RP24-78D24 BAC into EL250 electrocompetent *E. coli*[35] with a pulse of 1.8 kV in a 0,1 cm cuvette during 6 ms. Colonies that incorporated the BAC were selected with chloramphenicol resistance. Second, the insert was amplified by PCR from FRT-KanR-FRT-TFP plasmid with primers that contained homologous ends to the intended insertion site. We used Q5® High-Fidelity DNA Polymerase (NEB, #M0491) for the insert amplification, products were purified and then electroporated into EL250 *E. coli* carrying the BAC. The used primers were:

Forward: CTCTAGACTTGCTTCCCTTGCTGTGCCCCCTCCAGCAG ACAGCCACGACGGCTA GCAGATAACTGATCAG. Reverse: ACGGAGT CGTAGTCGAGGTCATAGTTCCTGTTGGTGAA GTTCACGTTGAGGGAT CTGAGTCCGGACTTG.

Recombination was induced by warming the culture at 42 °C for 15 min. After that, positive colonies were selected with double resistance to chloramphenicol and Kanamycin. Third, the FRT recombination was induced by adding L-arabinose at 0.2%, at 30 °C for 1 h. EL250 cells have a *Flpe* gene under the control of the arabinose-inducible promoter. Positive colonies were selected by persistence of chloramphenicol resistance and loss of kanamycin resistance. Once the TFP-Myc BAC was obtained, we amplified the recombination regions with PCR and sequenced the product (https://www.secugen.es/) to ensure the correct translational frame of the fusion protein. As a routine, all BAC-contained bacterial cultures were incubated at 30 °C.

## Mouse lines and transgenesis

We performed a 2-cell stage microinjection in the CNIC Transgenic Unit to generate the $Myc^{+/+}$, $Myc^{2TFP+}$ mouse line. Circularized TFP-Myc BAC was microinjected at a concentration of 0.675 ng/µl to C57BL/6JCrl 2-cell embryos. A total of 131 embryos were microinjected, 94 embryos were transferred to 4 recipient mothers and 11 animals were weaned. Animals with TFP-Myc BAC integration were crossed between them to establish the mouse line, homozygous mice are viable and show no obvious phenotypic alterations.

The $Myc^{GFP/GFP}$ reporter line has been previously described by Huang et al.[33]. The $Myc^{GFP/+}$,$Myc^{+/2TFP}$ mouse line was generated by crossing $Myc^{2TFP/2TFP}$ mice with $Myc^{GFP/GFP}$ mice.

The $Myc^{GFP\Delta2}$ and $Myc^{GFP\Delta7-3}$ alleles were generated by CRISPR-Cas9 as described below in CRISPR-Cas9 mutagenesis in mESCs. The Cas9-guide RNA mix was microinjected in 1-cell stage $Myc^{GFP/GFP}$ embryos in the CNIC Transgenesis Unit. Then, embryos were transferred to a foster mother and the progeny genotypes for the Enh7-3 or Enh-2 deletion. crRNA and primers used are listed in Supplementary Tables 3 and 4.

Wild-type mice of the CD1 strain were bred in-house and used for experiments not involving genetically modified mice. All specimens analyzed were very early embryos and sex was not determined.

## CNV analysis of $Myc^{2TFP/2TFP}$ mouse

Genomic DNA from the tail tissue of a direct descendant of the founder $Myc^{2TFP/+}$ mouse was used for sequencing. DNA was sequenced by Illumina NGS system using whole-genome sequencing method. We obtained on average 150 bp reads with a 20 read depth. CNVnator tool was used to analyze the genomic sequence (https://github.com/abyzovlab/CNVnator). The algorithm estimates Read Depth (RD) mean and variation in sliding windows. Windows of similar RD are merged into segments and the distribution of the RD across the segment compared to RD to the entire genome or surrounding regions. Two tests are reported, a $t$-test and a test based on modeling a Gaussian distribution for the RD. The output produces several columns indicating the type of alteration (deletion/duplication), the segment size, read depth of the segment normalized to 1, $t$-test for the distribution of the RD of the bins within the segments compared to all segments, probability of RD values within the region to be in the tails of a Gaussian distribution describing frequencies of RD values in bins, $t$-Test for the segment for the middle of the segment, probability of being in the tails of a Gaussian distribution for the middle of the segment and proportion of reads mapped with mapping quality == 0. For this analysis, we applied window sizes of 10 K and 20 K. The following filter was applied subsequently: Both $t$-Test (columns 5 and 7) < 0.05 and the ratio of mQ0 (column 9) < 0.01.

## Embryo retrieval

Midday of the day that the vaginal plug is detected is considered gestational day 0.5 (E0.5). Females were sacrificed by $CO_2$ inhalation on day 3, and the uterus was extracted. Embryo extraction at E3.5 was performed by flushing the blastocysts out of the uterus under a dissection scope using a 1 ml syringe with a 23-G needle. Blastocysts were fixed in paraformaldehyde (PFA) (Merk, #158127-500 G) 2% in PBS overnight at 4 °C. After fixation embryos were washed in PBS several times. For E5.5-6.5 embryo harvest, females were sacrificed by $CO_2$ inhalation at the midday of day 5 or 6 and the abdominal cavity was opened to expose the uterus, which was transferred to sterile PBS. Then, working in cold PBS under the scope, the muscular uterine wall was carefully ripped and then the decidual layer and the Reichert´s membrane were removed. Exposed embryos were fixed in PFA 2% in PBS overnight at 4 °C. After fixation embryos were washed in PBS several times. For the E9.5 embryo harvest, females were sacrificed by $CO_2$ inhalation at the midday of day 9 and uterus were extracted. The muscular uterine wall was ripped, and the yolk sack and the amnion

were removed. Embryos were then fixed in PFA 2% in PBS overnight at 4 °C. After fixation embryos were washed in PBS several times.

## Mouse ESC derivation and establishment

$Myc^{+/+}$,$Myc^{2TFP/2TFP}$ and $Myc^{GFP/+}$,$Myc^{+/2TFP}$ mouse lines were used for mESCs derivation. 2 mESC lines were derived from each mouse line. Blastocysts were collected and transferred individually to a 24-well plate containing a freshly inactivated MEF feeder layer. Cultures were maintained for 5–8 days in SR + LIF+2i medium (see below) without disturbance. When blastocysts attached, the inner cell mass grew as an individual colony. Then, each colony was trypsinized and individually passaged to 12-well plates. Each blastocyst gave rise to one cell line, which went through several passages before freezing.

Cultures were checked for regular karyotype (see below) at their establishment. Finally, ESCs were routinely maintained on mitoMycin-C-inactivated MEF feeder layers. Cells were passed every 2 days and frozen at 800,000–900,000 cells per vial. ESCs were routinely cultured in an SR medium (see below).

## Karyotyping

Approximately $1.5 \times 10^6$ cells in a 60 mm plate were treated with 0.5 µg/ml KaryoMAX™ Colcemid™ Solution in PBS (ThermoFisher, #15212012) for 1 h at 37 °C. Cells were washed with PBS, trypsinized, and resuspended in 6 ml of a hypotonic solution (75 mM KCl) for 7 min at room temperature (RT). 8 drops of a fixative solution (3:1 methanol/glacial acetic acid) were added and then centrifuged for 5 min at 225 g at RT. The supernatant was removed and 6 ml of the fixative solution was added and incubated 20 min in ice. This last step was repeated twice, and 2 ml of the sample was kept at − 20 °C for later use. Finally, a small volume of the solution was dropped on a slice, dried, and mounted with DAPI (PALEX, #416399 H-1200). The karyotype of a cell line was considered suitable when more than 80% of the cells exhibited normal karyotype (40 chromosomes).

## Mouse ESC in SR + LIF medium

0.5–0.8 million of mESCs were cultured for 2 days without MEFs in SR + LIF culture medium, which contained High glucose DMEM (ThermoFisher, #11965092), 1% Pyruvate (ThermoFisher, #11360070), 15% KO-SR, 2× LIF, 1% nonessential amino acids (100×), and 1% Penicillin/Streptomycin (Pen/Strep) (10,000 U/ml; 100×), 0,1% 2-beta-mercaptoethanol (50 mM). For confocal analysis, 0.5 million cells were seeded on Human fibronectin (VWR, #734-0085) coated plates with glass-bottom (MatTek, #•P35G-1.5-14-C) for 2–4 days. For flow cytometry analysis, 0.02 million cells were seeded on 0,1% gelatin-coated 24-well plates for 1–5 days.

## Mouse ESC in SR + LIF + 2i medium

0.5–0.8 million mESCs were cultured as described above with SR + LIF culture medium supplemented with 3 µM CHIR99021 and 1 µM PD0325901 (2i). For confocal analysis, 0.5 million cells were seeded in fibronectin-coated plates with glass-bottom for 2–4 days. For flow cytometry analysis, 0.02 million cells were seeded in 0.1% gelatin-coated 24-well plates for 1–5 days.

## Mouse ESC in N2B27 + 2i medium

0.5–0.8 million ESCs were cultured for 2 days in SR + LIF+2i medium (see above) in 0.1% gelatin-coated 35 mm plates. Then, cells were detached, and 0.5 million cells were cultured for 2 days in N2B27 medium supplemented with 1× LIF and 2i in 0.1% gelatin-coated 35 mm plates. N2B27 medium consisted in DMEM/F12 50% (Gibco, #11330−032), Neurobasal media 50% (Gibco, #21103−049), 2 mM L-glutamine (Gibco, #25030), 0.1 mM β-Mercaptoethanol, 1× N2 supplement (Invitrogen, #17502048), 1× B27 supplement (Invitrogen, #17504044) and 1× Pen/strep. For flow cytometry analysis, 0.02 million cells were seeded in 0,1% gelatin-coated 24-well plates for 2 days.

## EpiLC culture in N2B27 + F/A medium

ESCs were converted into EpiLCs following the protocol described by Hayashi and colleagues[7]. 0.5–0.8 million ESCs were cultured for 2 days in SR + LIF+2i medium (see above) in 0.1% gelatin-coated 35 mm plates. Then, cells were detached, and 0.5 million cells were cultured for 2 days in N2B27 medium supplemented with 2× LIF and 2i in 0.1% gelatin-coated 35 mm plates. N2B27 medium consisted in DMEM/F12 50% (Gibco, #11330–032), Neurobasal media 50% (Gibco, #21103–049), 2mM L-glutamine (Gibco, #25030), 0.1 mM β-Mercaptoethanol, 1× N2 supplement (Invitrogen, #17502048), 1× B27 supplement (Invitrogen, #17504044) and 1× Pen/strep.

For confocal analysis, 0.5 million cells were passaged and cultured for a further 2 days in N2B27 medium supplemented with 20 ng/ml activin A (R&D Systems, #338-AC-050/CF), 10 ng/ml FGF2 (R&D Systems, # 233-FB-025) and 1:100 KO-SR in fibronectin-coated plates with glass-bottom for 2 days. For flow cytometry analysis, 0.02 million cells were seeded in fibronectin-coated 24-well plates in N2B27 medium supplemented with 20 ng/ml activin A, 10 ng/ml FGF2, and 1:100 SR for 2–5 days.

## Western blotting

5 million cells were trypsinized and washed twice in PBS. Cell pellets were lysed in RIPA buffer (Tris 50 mM, NaCl 150 mM, Triton X-100 1%, EDTA 1 mM, 2,5 g NaDOC, and add distilled $H_2O$ to 100 ml final volume) with cOmplete™, Mini, EDTA-free Protease Inhibitor Cocktail (PIC) (Merk, #4693159001). Protein concentration was measured with the BCA protein Assay kit (ThermoFisher, #23227). 40 μg of samples were denaturalized and loaded in 8% agarose gel and transferred to an Immuno-Blot PVDF Membrane (Bio-rad, #1620177) with 0.2 μm pore size. After transfer, the membranes were blocked with 5% milk in 0.1% PBS-tween for 2 h. Primary antibodies: MYC (rabbit, Abcam, #ab32072), GFP (rabbit, Takara, #632593), and Vinculin (mouse, Merk, #V4505-100UL) were used at 1:1000 and incubated at 4 °C o/n. Then, the membranes were washed with 0,1% PBS-tween. Anti-rabbit-HRP (Dako, #0448) and anti-mouse-HRP (Dako, #0447) secondary antibodies were used at 1:500 and incubated for 1 h. After that, membranes were washed at least 5 times and incubated for 1–2 min with Immobilon Forte Western HRP substrate (Merk, #WBLUF0100). Finally, membranes were revealed in iBright Imaging Systems.

## Circularized Chromosome Conformation Capture (4C)-seq

5 million cells were used for each sample and replicate. When the culture reached 70–80% confluency, cells were detached with trypsin 0.25% and then fixed with fresh 2% PFA. Then, the cell pellet was lysed with 50 mM Tris pH 7.5, 150 mM NaCl, 5 mM EDTA, 0.5% v/v NP40, 1.15% v/v Triton X-100, and 1× PIC. After this, we proceeded to chromatin digestion with Csp6I (CviQI) (ThermoFisher, #ER0211 or NEB, #R0639L). Digestion efficiency was checked by gel electrophoresis. Digestion products were directly used for the first ligation with T4 DNA ligase. After ligation, DNA was purified using phenol: chloroform and precipitated with 3 M NaAC, pH 5.2 (1/10× vol.), glycogen (0.05–1 μg/μl final concentration) and 100% EtOH (~2.5× vol.). Then, we proceeded to the second digestion and ligation for library preparation. We used NlaIII (NEB, #R0125L) for the second digestion and digestion products were directly used for the ligation step. This second ligation was done in a large volume, to favor intra-molecular ligation events. After this, the final products were purified with AMpure XP beads on a magnet (Bechman Coulter, #A63881). The DNA obtained was used for library generation using 2 PCR rounds (Primers are listed in Supplementary Table 2). In PCR 1, we designed primers aligned to specific sequences of the viewpoint and adapters for the second PCR. This PCR was amplified with an Expand long template system (Roche/merk, #11681834001). Then the product was purified with AMpure beads and used as a template for PCR 2. For the second PCR, we designed primers with P5 sequence for the forward (common for all the samples), and the reverse primers have P7 sequence and index

sequences, different for each sample. This PCR was amplified with NEBNext High-fidelity 2× PCR master mix (NEB, #M0541S). Finally, products were purified with Ampure beads and sequenced by Illumina sequencing technology in the CNIC Genomic Unit. The size of the libraries was checked using the Agilent 2100 Bioanalyzer and the concentration was determined using the Qubit® fluorometer (Life Technologies). Libraries were sequenced on a NextSeq2000 (Illumina) to generate 60 × 42 bases paired-end reads. FastQ files for each sample were obtained using bcl2fastq 2.20 Software (Illumina).

Sequences were then processed using the 4Cpipe pipeline (https://doi.org/10.1016/j.ymeth.2019.07.014) over mm10 reference genome with Csp6I and NlaIII as first and second cutter, a viewpoint at position chr15: 61985921 (TGGGGCACAAGCTGGAGTAC), a max mapping mismatch of 2, a normalization factor of 10,000 with the top 10 fragments removed, a read quality trimming cutoff of 10 and a running mean window of 21 fragments. The rest of the parameters are default. The data displayed has been summarized using the median over 4 replicates per condition.

## TFP-Myc BAC FISH in mESCs

$Myc^{G/G}$ and $Myc^{+/+}$,$Myc^{2TFP/2TFP}$ mESC were cultured in SR + LIF condition and fixed as described in the Karyotyping protocol. Then the samples were transferred to the Cytogenetic Unit at CNIO (sro-driguezp@cnio.es). Three probes were designed targeted to chromosome 15, chromosome 19, or the BAC RP24-78D24 (chr15). The control probes that target chromosome 15 were generated using digested BAC RP23-80F2, localized in the 15qA1 region of the chromosome and tagged with a blue fluorophore. The probe targeted to chromosome 19 was generated using digested BAC RP23-353A20, which contains DNA of the 19qA1 region and was tagged with a green fluorophore. Lastly, BAC RP24-78D24 (15qD1) digested fragments were used to generate probes tagged with red fluorophore. BAC DNAs were directly labeled using a "Nick translation kit" according to the manufacturer's specifications.

Glass slides with nuclei were incubated for 10 min at 90 °C and dehydrated in ethanol series 70–80% and 100% for 3 min each step. A hybridization mix was prepared by mixing the probe and the hybridization buffer. Then, the glass slide and coverslip were sealed and incubated in the DAKO hybridizer machine following the manufacturer's instructions. After that, the slides were washed with PBD and incubated for 2 min in 0.4× SSC buffer (Sigma, #1002100191) and 0.3% NP40 at 78 °C. Then the slides were put in 2× SSC and 0.1% NP40 for 5 min at RT and dehydrated in alcohol series as described before. Finally, samples were mounted with Vectashied with DAPI.

FISH images were captured using Leica Microsystems CMS GmbH, DM5500B or CCD camera (CV-M4 + CL Mega Pixel Progressive Scan camera) connected to a PC running the Cytovision v 7.4.0.0 image analysis system (Applied Imaging Ltd., UK) with focus motor and Z stack software.

## CRISPR-Cas9 mutagenesis in mESCs

Clusters and putative enhancer regions were deleted by the CRISPR-Cas9 system. crRNA guides were designed using CRISPOR (crispor.tefor.net/). 2 pairs of crRNAs (IDT) were used for each deletion. crRNAs were mixed with Alt-R® CRISPR-Cas9 tracrRNA (IDT, #1072532) in an equimolar way, and combined with Cas9 (provided by Pluripotent Cell Technology Facility at CNIC) or Cas9 HiFi from IDT (Alt-R™ S.p. HiFi Cas9 Nuclease V3 #1081060) to make the RNP complex and electroporated using 1400 V, 3 pulse × 10 ms in the Invitrogen Neon electroporator system. Cells were transfected in parallel with a plasmid expressing a fluorescent reporter. After 1–2 days of culture, cells positive for the reporter were separated using a fluorescent cell sorter (BD FACSAriaTM II). Then single colonies were picked, expanded, and analyzed by PCR of the deletion. crRNA and primers used are listed in Supplementary Tables 3 and 4. TF binding sites were mutated by

CRISPR-Cas9 system as described above but only one crRNA was used for each binding site. Positive cells were analyzed by PCR and subsequent enzyme restriction. crRNA, primers, and restriction enzymes used are listed in Supplementary Tables 3 and 4.

## Co-culture and cell competition assays
Cells were cultured under SR + LIF or N2B27 + F/A culture conditions. Cells were mixed for co-cultures with 0.02 million cells at a 1:1 ratio in 24-well plates and were maintained for 5 days in the mentioned mediums. The percentage and evolution of each population were followed daily by passing the co-cultures in a flow cytometer for 5 days, based on the fluorescent tag of the ubiquitously expressed PGKtdTomato in one of the cell populations. On the day of analysis, cells were trypsinized and cell suspensions were analyzed in BD LSRFortessaTM Special Order Research Product (laser wavelengths 405, 488, 561, 640). DAPI was used for identifying dead cells.

## Whole-mount embryo immunofluorescence
For embryos, the same procedure was followed for all the stages and staining procedures. Embryos were permeabilized in 0.5% Triton X-100 for 30 min. Excess triton was washed in PBS for 15 min. Then, embryos were blocked with a TNB-blocking reagent (Perkin Elmer, FP1012-X) for 1 h. Primary antibodies were incubated overnight at 4 °C. Afterward, embryos were washed several times with 0.1% Triton X-100 and incubated with secondary antibodies and DAPI 1 h at room temperatures. After secondary antibodies incubation, embryos were washed with 0.1% Triton X-100 at least 5 times. Primary and secondary antibodies were diluted in the blocking solution. Primary antibodies: GFP Goat Polyclonal Antibody (Goat, 1:200, OriGene, #R1091AP), Human/Mouse Brachyury Antibody (mouse, 1:250, R&D systems, #AF2085-SP), and Anti-SOX2 antibody (rabbit, 1:200, Abcam, #ab97959). Secondary antibodies: 648 donkey anti-goat (1:500, Life Tech, #A-21447), 488 goat anti-mouse (1:500, Life Tech, #A32723) and 633 goat anti-rabbit (1:500, Life Tech, #A-21070).

## Mouse ESC immunofluorescence
For mouse ESCs, a similar protocol was followed for all staining procedures. Fixed samples were permeabilized with 0.25% Triton X-100 for 15 min. Excess Triton was washed in PBS and blocked with TNB for one hour. The primary antibodies were incubated at 4 °C o/n. After several washes with 0.1% Triton X-100, secondary antibodies and DAPI were incubated for 1 h at room temperature. Cells were then washed at least 5 times with 0.1% Triton X-100 and mounted in Vectashield mounting medium. To analyze GFP-MYC and TFP-MYC endogenous fluorescence, cells were fixed, permeabilized, and incubated for 1 h only with DAPI at RT. Primary antibodies were: phospho-p44/42 MAPK (Erk1/2) (Thr202/Tyr204) (rabbit, 1:200, Cell Signaling, #4370 #9106), Anti-SOX2 antibody (rabbit, 1:200, Abcam, #ab97959) and Oct-3/4 Antibody (C-10) (mouse, 1:200, Santa Cruz, #sc-5279). Secondary antibodies: 633 goat anti-rabbit (1:500, Life Tech, #A-21070) and 568 goat anti-mouse (1:500, Life Tech, #A-11004).

## Confocal microscopy
Leica TCS SP8 coupled to a DMi8 inverted confocal microscope Navigator module equipped with white light laser was used for imaging. A ×40 oil objective and 1024 × 1024 pixels, A.U. set to 1 were commonly used for mESCs immunostaining and embryos from E3.5 to E6.5. A 20× glycerol objective and 1024 × 1024 pixels, A.U. set to 1 were commonly used E9.5 embryos.

## 3D analysis of posterior embryonic bud region at E9.5 embryos
The same confocal equipment as in the previous section was used for this acquisition. Embryo tails were mounted in ProLong™ Gold Antifade Mountant (ThermoFisher, #P10144) with a cover glass N° 0 at both sides. The 2 sides of the posterior bud were acquired separately with ×20 glycerol objective, 1024 × 1024 pixels, and optimal Z. Then the 3D reconstructions were made manually with the BigStitcher plugin in ImageJ. 3D representations were made with Imaris Microscopy Image Analysis Software.

## Nuclear and cytoplasmic signal detection
Confocal images were analyzed with ImageJ (http://imagej.nih.gov/ij). To quantify nuclear signal in cells and E6.5 embryos, nuclei were detected by DAPI staining and segmented by using a threshold tool to create a mask. Manual correction was applied to ensure the segmented objects belonged to individual cells. Masks were applied to the corresponding channel and a measurement tool gave the mean intensity for every single object. For cytoplasm signal quantifications, masks were obtained by subtracting the DAPI mask from the whole cell mask, which was created thanks to the Tomato membrane marker. Signal intensity measuring was performed as in the nuclei. For quantification of the Myc signal in blastocysts, the fluorescence intensity was determined by quantifying several sections for each nucleus. Alternatively, we segmented the nuclei using Cellpose[66] in 3D on the MYC channel. Nuclei were segmented in the 3D space and a mask was created from the segmentation. To quantify the intensity of each nucleus, we used 3D Intensity Measurements from the FIJI MorphoLibJ package[67] quantifying the raw MYC signal in each masked element.

## Statistical analysis and reproducibility
Comparisons and graphs were made with GraphPad Prism 5.0a. Statistical analyses have been included in the Figure legends and include corrections for multiple comparisons when applicable. Nonparametric tests were used for all comparisons. 1-way ANOVA was used for multiple comparisons and 2-way ANOVA for multiple comparisons with temporal analysis. No statistical method was used to predetermine the sample size. No data were excluded from the analyses. Randomization is not applicable to the experiments performed. No blinding was used during experiments and outcome assessment.

## Reporting summary
Further information on research design is available in the Nature Portfolio Reporting Summary linked to this article.

## Data availability
The Raw data generated in this study have been deposited in the Figshare database and are available from https://doi.org/10.6084/m9.figshare.24794559. The data for some images that were not quantified but only qualitatively analyzed and were too large for the allowance in the Figshare repository can be obtained from M.T. upon request. Sequencing data generated in this study are available from the GEO database with accession number GSE222299. The ChIPseq data used in this study are publicly available from the ChIPseq Atlas (https://chip-atlas.org/peak) and the PLAC-seq data used in this study are available from the 3D genome database (http://3dgenome.fsm.northwestern.edu/). Source data are provided with this paper.

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

## Acknowledgements

We thank members of the Torres laboratory for fruitful discussions and advice on this work. We thank M. Sendra for help in dissecting mouse embryos, C. Torroja from the CNIC Bioinformatics Unit for sequence analysis, Cristina Álvarez Diago from the CNIC Pluripotent Cell Technology for generation and analysis of mutant mESCs, M. Franke from CABD and Raquel Ruoco from University of Geneva for advice on the 4C protocol, Rui Benedito from CNIC for BAC recombineering protocols, the CNIC transgenesis unit for generation of mouse lines, the CNIC Genomics Unit for sequencing and the CNIC Advanced Microscopy Unit for advice on confocal microscopy. Funding: Grants PGC2018-096486-B-I00 and PID2022-140058NB-C31 from the Agencia Estatal de Investigación to M.T.; European Commission H2020 Program grant SC1-BHC-07-2019. Ref. 874764 "REANIMA" to M.T. Comunidad de Madrid grant P2022/BMD-7245 CARDIOBOOST-CM to M.T. The CNIC is supported by the Instituto de Salud Carlos III (ISCIII), the Ministerio de Ciencia, Innovación y Universidades (MICIU) and the Pro CNIC Foundation, and is a Severo Ochoa Center of Excellence (grant CEX2020-001041-S funded by MICIU/AEI/10.13039/501100011033).

## Author contributions

Conception and design of the work: M.T. and L.L.; Data acquisition: M.T., L.L., C.D., R.S., S.T., E.S., G.G., F.J.M., S.R.; Data analysis: M.T., L.L., C.D., A.R., T.B., M.R.; Data interpretation: M.T., L.L., C.D., F.S., A.R., T.B.; Production of experimental materials: G.G., E.S, F.S.; Manuscript draft: M.T., L.L.; Manuscript revision: M.T., L.L., C.D., F.S., A.R.

## Competing interests

The authors declare no competing interests.
