## [Peer Review File · Nature Communications]

REVIEWER COMMENTS

Reviewer #1 (Remarks to the Author):

In this manuscript, the authors reported the identification and characterization of a novel regulatory elements of *Myc* that direct the expression in pluripotent cell populations of mouse embryos at pre- and early post-implantation stages. *Myc* has multiple important roles in pluripotent cells as the authors discussed in the introduction part, but the regulatory elements to direct the expression in these cell populations remain unclear. The authors applied the large-scale dissection of the flanking region of *Myc* and found the Early Embryonic Expression (EEE) topologically-associated region. Further dissection revealed the separately-regulated enhancers activated in naïve and formative pluripotent stem cells. Informatic analyses identified multiple transcription factors that may involve in the activation of these enhancers.

The authors aim to analyze the regulatory element of *Myc* in comprehensive manner. Their plan has been accomplished to some degree, but not completely for the following reasons in the comment. In addition, the functional analysis of the identified enhancers is immature because they didn't identify the transcription factors that are responsible to activate them. Therefore, this paper has not reached the level of acceptance for publication in Nature Communications.

1. Figure 1i: Why do the authors choose *Foxd3* as the ES-cell transcriptional regulator? According to the previous Chip-seq analyses, the core pluripotency-associated transcription factors *Pou5f1*, *Sox* and *Nanog* also occupy this ~210kb region as the authors mentioned in Fig 4g.

2. Page 7 middle: Does 'FP-MYC expression levels' mean TFP-MYC and GFP-MYC expression levels?

3. The authors stated that if sub-clusters A and D contained major enhancer activity in these culture conditions, it would have been detected based on the analysis of heterozygous deletion of sub-cluster C. However, homozygous deletion of enhancer-2 in sub-cluster A region results significant reduction in 2i condition. That argument fails logically. Moreover, this discrepancy spoils the comprehensiveness of their genome analysis.

4. The authors apply the culture medium with serum, LIF and 2i (MEK inhibitor and GSK inhibitor) as '2i'. However, the canonical 2i culture is serum-free in which the self-renewal of ES cells depends to 2i. The authors should not state their culture condition as '2i'. In addition, in the comparison between serum+LIF and Serum+LIF+2i, the differential gene expression may be caused by the response against

the signal modulation rather than the different cell state. What happens on the expression of Myc in serum+LIF+1i (MEKi or GSKi)?

5. Fine dissection of the regulatory elements to identify the transcription factors responsible for their activation will be required.

Reviewer #2 (Remarks to the Author):

The manuscript by Li-Bao et al. dissects the regulatory elements of Myc gene in pluripotent stem cells in vitro and during early developing mouse embryos. They identified an early embryonic expression (EEE) region that contains the enhancers to drive Myc expression in different pluripotent states during pre- and post-implantation development. Notably, they identified a formative pluripotency related enhancer dedicated to Myc expression in post-implantation epiblast. Deletion of this enhancer dampens the competitive ability of formative and primed pluripotent cells.

The main finding of this study is the identification of a topologically defined enhancer cluster dedicated to early embryonic expression and uncover a modular mechanism for the regulation of Myc expression in different states of pluripotency. While some new information was generated regarding the transcriptional control of Myc expression from this study, the data presentation is not of highest rigor/quality and overall conceptual novelty is limited, which makes it a borderline candidate for publication in Nature Communications.

Main critiques:

1. The sentence “Each of these in vitro states can be maintained stably in culture by using specific culture conditions⁵.” is not an accurate statement. The authors recognized that “The epiLCs state can also be transiently obtained in vitro using conditions that lead to the EpiSC state after a few days in culture⁷” which could be more accurate or specific, that is, only 48 h of F/A after exit of naïve pluripotency can be considered a formative state.

2. In Fig. 1, it is very confusing to understanding those rearrangement fragments with solid border and dash borders, what do they mean? The term “InvMyc1” is not clear. Is it a deletion starting from downstream of Pvt1? Likely not. If it is an inversion, what is the orientation of the fragment? Did the authors perform staining of negative control embryos (i.e., MycDel3MB^{-/-}) to eliminate the background of Myc staining?

3. In Fig. 3, the authors failed to obtain sub-cluster A and D homo deletion clones. Here more details are needed when performing the CRISPR experiment. If more than 100 clones were tested but still cannot get homo deletion, it will suggest an essential role of sub-cluster A and/or D in maintaining the Myc expression and cell proliferation. Since the hetero A or D deletion will likely has no phenotype, the authors should emphasize that the functions of sub-cluster A or D on Myc expression is nonconclusive.

4. In Fig. 3g', why are there few dots with high variance in naïve pluripotency data? Same in Fig. 4e', the data has few dots but high variance. Both experiments need to be repeated.

5. In Supplemental Figure 4, please show WB data with both 65KD and 95KD region.

Minor points:

1. Fig. 2h'', the h'' cannot be seen in the panel.

2. "We also analyzed FP-MYC expression levels". What does FP-MYC mean?

3. The authors are advised to avoid using unconventional abbreviation words, such as epiLCs, epiSCs (should have capital "E"), CHIP (should be "ChIP"), and transcriptional start (should be "transcriptional start site").

Reviewer #3 (Remarks to the Author):

In this manuscript, Li-Bao et al. interrogate the enhancer driven transcriptional regulation of the well-studied MYC gene in several different contexts including, two different pluripotent states in vitro and during early development in vivo. By knocking-in a BAC clone carrying 250kb around a MYC-TFP reporter gene and comparing the expression of this transgene to the endogenous MYC-GFP during development, they argue that this 250kb region is sufficient to recapitulate MYC regulation in early pluripotent and multipotent cells. In contrast, knock-in MYC-TFP reporter diverges from the endogenous MYC-GFP expression in later embryonic tissues, suggesting that additional regulatory elements beyond the 250kb region are required for proper regulation later in development. Then the researchers focus on individual candidate enhancers within the 250kb window, and identify two regions that when deleted show cell-state specific enhancer contribution to MYC expression. One of these regions has also been deleted in developing embryos, providing more robust evidence of its effect. The researchers use the logic that MYC expression is a readout for competitive fitness for selection of the "best" developmental cells. In

support to this notion, deletions of specific MYC enhancers result into a negative selection, where cells that carry the deletion are outcompeted from the population.

Overall, the work is nicely presented and the data well-organized. The experiments done provide clear evidence that two cell-type specific enhancers (one for the naïve and one for the formative state) play a role in MYC regulation. However, there are several weaknesses on the narrative, the description and the justification of their approach and the conclusions that they draw from their data that need to be addressed. Below are my main concerns and suggestions:

Main Concerns:

- Selection and characterization of the candidate enhancer regions.
 - o Besides the genomic tracks of some ChIP-seq data in Figure 1i and the claim for “meta-analysis”, there is no clear explanation of the criteria used for the selection of top candidate enhancers. Is it the relative strength of each of these signals? Any prioritization based on the type of TFs bound? As a reader I assumed that there was some integration of the deletions shown in 1A with the ChIP tracks shown in 1i. However, the deletions/perturbations shown in A are so large I am not sure what they add to the selection criteria.
 - o The authors also present the MYC TAD that might eliminate interactions beyond its boundaries, they do not dig deeper into the actual insulation or relative contact frequencies with the putative enhancers. Given that a large number of high-resolution 3D assays (e.g micro-C, promoter Capture or HiChIP) are publically available both in mouse ESCs – and even EpiLCs (Capture-C), it is a wasted opportunity not to utilize these data to associate/predict regulatory impact in a cell-type specific manner. This would strengthen the novelty and relevance of the study.
 - o Moreover, the bioinformatics analysis of the candidate enhancers is very limited, simply listing the potentially relevant TFs that are bound there. Are these regions more enriched in binding or combinatorial bindings compared to other putative regulatory regions within the TAD? An easy follow up, would be to introduce point mutations or deletions of specific binding motifs to corroborate the relevance of one or more TF binding sites on the enhancer activity.

- The establishment of a TFP-MYC transgene is a potentially useful tool that provided interesting insights, but with unclear relevance for this manuscript. Their experiments indeed support that the BAC clone likely contains all critical regulatory elements for MYC regulation in early development -although an actual quantitation and comparison of the MYC-TFP and endogenous MYC-GFP expressions is needed in the main figure (listing number of embryos and cells quantified). In contrast, their conclusion that regulatory elements required for later stages are likely located further away is not well-supported. The fact that the BAC is out of context might have some suppressive effects on the transgene, even if all important cell-type-specific enhancers are present. Another comment regarding the MYC-TFP experiment is that it is unclear how the introduction of this tool so early in the manuscript contributes to the follow up experiments shown after. Does the characterization of this reporter tool fit into the overall

scope of the rest of the manuscript? What did the characterization of the MYC-TFP add to the overall criteria for choosing enhancer regions for deletion? It seems that the selection happened already based on Figure 1i.

- The claim that MYC-enhancer deletions reduce fitness is very bold in context of the experiments done. Cell growth and differentiation experiments were done to show that upon MYC-enhancer deletion some cells are now less “fit” than others. The authors relate this process to selection of the best pluripotent cells in the embryo. However, all these experiments could simply indicate that manipulating MYC has an overall effect on proliferation, which is expected. Do they claim something beyond the obvious? Maybe it would be more convincing to demonstrate “fitness” in a developmental context. Would they contribute to embryos more?

Minor Concerns:

- o Authors describe the SR+LIF condition as “mixed pluripotency”, by which I assume they mean a mixture of naïve and formative pluripotent cells. Where do they base this description? The paper they cite is not relevant.

- o How do any of the reported deletions affect PVT1 gene? There are various reports showing competition between MYC and PVT1. It is only mentioned in the discussion “These results suggest that neither the long-noncoding RNA encoded by Pvt1, nor its transcriptional activity regulate Myc expression in ES cells”, but I could not find the data supporting this claim.

- o All imaging data should be quantified listing the numbers of embryos and cells used.

- o In Figure F4G the authors do not describe the image and the TF enrichment adequately. What do they consider binding and what do these lines represent (# of reads)? Do they consider FOXD3 as a binding factor at enhancer-2 based on 2 lines?

- o 4C analysis is missing since there is no mention of the algorithms and steps used to generate supplementary figure 2a and 2b.

- o In page 8 it is written: “While we could not obtain sub-clusters A and D homozygous deletions, sub-cluster C deletion in heterozygosity provokes detectable MYC level reduction in SR+LIF and F/A conditions (Supplementary Figure 7a-e), suggesting that if sub-clusters A and D contained major enhancer activity in these culture conditions, it would have been detected.” A better explanation should be provided.

- o Figure 5 is presented before figure 4 in the manuscript and somehow this complicates the flow.

Reviewer #1 (Remarks to the Author):

In this manuscript, the authors reported the identification and characterization of a novel regulatory elements of Myc that direct the expression in pluripotent cell populations of mouse embryos at pre- and early post-implantation stages. Myc has multiple important roles in pluripotent cells as the authors discussed in the introduction part, but the regulatory elements to direct the expression in these cell populations remain unclear. The authors applied the large-scale dissection of the flanking region of Myc and found the Early Embryonic Expression (EEE) topologically-associated region. Further dissection revealed the separately-regulated enhancers activated in naïve and formative pluripotent stem cells. Informatic analyses identified multiple transcription factors that may involve in the activation of these enhancers.

The authors aim to analyze the regulatory element of Myc in comprehensive manner. Their plan has been accomplished to some degree, but not completely for the following reasons in the comment. In addition, the functional analysis of the identified enhancers is immature because they didn't identify the transcription factors that are responsible to activate them. Therefore, this paper has not reached the level of acceptance for publication in Nature Communications.

We thank the reviewer for the comments and hope to have increased the level of the manuscript with the new experiments

1. Figure 1i: Why do the authors choose Foxd3 as the ES-cell transcriptional regulator? According to the previous Chip-seq analyses, the core pluripotency-associated transcription factors Pou5f1, Sox and Nanog also occupy this ~210kb region as the authors mentioned in Fig 4g.

As suggested by the reviewer, we have now included Oct4, Nanog and Sox2 tracks in Figure 1 and other Figures in which this is relevant.

2. Page 7 middle: Does 'FP-MYC expression levels' mean TFP-MYC and GFP-MYC expression levels?

Yes indeed. We have now corrected the text to make it explicit.

3. The authors stated that if sub-clusters A and D contained major enhancer activity in these culture conditions, it would have been detected based on the analysis of heterozygous deletion of sub-cluster C. However, homozygous deletion of enhancer-2 in sub-cluster A region results significant reduction in 2i condition. That argument fails logically. Moreover, this discrepancy spoils the comprehensiveness of their genome analysis.

In our opinion the reviewer is partially correct, given that we were referring specifically to the FA culture conditions and to the putative detection of enhancers with activity equal or above that of subcluster C in these specific culture conditions. Indeed, the argument only applies to the detection of regulatory elements of strength at least similar to the one contained in subcluster C, so the argument can only be sustained for this case and for the FA culture conditions, whereas we cannot exclude the presence of other elements with activity below that of subcluster C. We have now rephrased this sentence to eliminate this incongruity.

4. The authors apply the culture medium with serum, LIF and 2i (MEK inhibitor and GSK inhibitor) as '2i'. However, the canonical 2i culture is serum-free in which the self-renewal of ES cells depends to 2i. The authors should not state their culture condition as '2i'. In addition, in the comparison between serum+LIF and Serum+LIF+2i, the differential gene expression may be caused by the response against the signal modulation rather than the different cell state.

The reviewer is correct and we now refer to the serum replacement+LIF+2i condition as Pro-naive instead of naive.

What happens on the expression of Myc in serum+LIF+1i (MEKi or GSKi)?

We have now tested these conditions suggested by the reviewer and found that PD does not change GFP-Myc expression alone, whereas Chiron activated Myc expression. These results show that the effect of the 2 inhibitors cannot be simply deduced from the actions of each single inhibitor in isolation and therefore the

interaction between the two inhibitors is complex and we feel this aspect goes beyond the scope of this manuscript. We include these results here for the appreciation of the reviewer.

5. Fine dissection of the regulatory elements to identify the transcription factors responsible for their activation will be required.

We have now identified putative TF binding sites in the enhancers characterized by a meta-analysis of previously published ChIP-seq experiments and by detection of DNA motifs susceptible to be bound by the candidate TFs around the center of the ChIPseq peaks identified. We then mutated the DNA sequences containing the DNA binding motifs by CRISPR-Cas9 deletion and characterized by cytometry the expression of the mutant lines in the different culture conditions already used in our previous submission and in N2B27+2i condition. Our results show the requirement for KLF5, ESRRB and FoxD3 binding sites for the full activity of enhancer 2 in all culture conditions except in FA, where KLF5 and FoxD3 sites are completely dispensable while the ESRRB site has a less important role. This result shows that transcription factors ESRRB, KLF5 and Foxd3, previously associated to naive pluripotency, regulate enhancer 2 for promotion of Myc expression in ESC cells. These new data were included in the new Figure 6 and new Supplementary Figure 9.

We used a similar approach for the characterization of enhancer 7-3 and identified a unique region engaged in binding transcription factors that regulate pluripotency. Interestingly, Otx2 binds this region preferentially in EpiSCs. Otx2 is a transcription factor activated in formative pluripotency epiblast cells and required for transition into the primed state. No other priming-related transcription factors ChIPseq peaks were reported for this region. Otx2 DNA binding sites are ill-defined, and only a very generic homeodomain binding site has been described. Unfortunately, we could not detect any Otx2 binding motif in the area bound by Otx2 in ChIPseq. We therefore chose to characterize a relatively large deletion (32 bp) centered around the middle of the Otx2 ChIPseq peaks. This deletion indeed affected GFP-Myc expression in FA culture conditions, however it also reduced the expression in the other conditions and even to a larger extent. In summary, although we have been able to find a role for a putative Otx2 binding site in regulating 7-3 activity, we failed to identify the basis for the specificity of this enhancer in the formative pluripotency condition. The fine characterization of this enhancer will therefore require further analyses and a systematic functional dissection of its sequences.

Further to this, and although not specifically requested by the reviewers, we completed the in vivo analysis by generating mice with enhancer 2 deleted. The in vivo results nicely recapitulate the in vitro discoveries showing that enhancer 2 specifically affects GFP-Myc expression in the epiblast of the blastocyst but not in the epiblast of post-implantation embryos. These data, together with the previously included results of enhancer 7-3 deletion in vivo, nicely show the complementarity of these two enhancers in the regulation of Myc expression during pluripotency progression in the mouse embryo. The new data are included in Figure 5 and Supplementary figure 8.

Reviewer #2 (Remarks to the Author):

The manuscript by Li-Bao et al. dissects the regulatory elements of Myc gene in pluripotent stem cells in vitro and during early developing mouse embryos. They identified an early embryonic expression (EEE) region that contains the enhancers to drive Myc expression in different pluripotent states during pre- and post-implantation development. Notably, they identified a formative pluripotency related enhancer dedicated to Myc expression in post-implantation epiblast. Deletion of this enhancer dampens the competitive ability of formative and primed pluripotent cells.

The main finding of this study is the identification of a topologically defined enhancer cluster dedicated to early embryonic expression and uncover a modular mechanism for the regulation of Myc expression in different states of pluripotency. While some new information was generated regarding the transcriptional control of Myc expression from this study, the data presentation is not of highest rigor/quality and overall conceptual novelty is limited, which

makes it a borderline candidate for publication in Nature Communications.

We thank the reviewer for the comments and hope to have addressed the main concerns

Main

critiques:

1. The sentence “Each of these in vitro states can be maintained stably in culture by using specific culture conditions⁵.” is not an accurate statement. The authors recognized that “The epiLCs state can also be transiently obtained in vitro using conditions that lead to the EpiSC state after a few days in culture⁷” which could be more accurate or specific, that is, only 48 h of F/A after exit of naïve pluripotency can be considered a formative state.

We have now modified this statement as suggested by the reviewer

2. In Fig. 1, it is very confusing to understanding those rearrangement fragments with solid border and dash borders, what do they mean? The term “InvMyc1” is not clear. Is it a deletion starting from downstream of Pvt1? Likely not. If it is an inversion, what is the orientation of the fragment? Did the authors perform staining of negative control embryos (i.e., MycDel3MB^{-/-}) to eliminate the background of Myc staining?

All represented segments are deletions except InvMyc1, which is an inversion. We now symbolized more specifically this inversion in Figure 1 and further explained the details in the Figure legend. Regarding the control for Myc expression, we have extensively used this immunofluorescence in E6.5 embryos (see Claveria et al, Nature 2013) and did not find any non-specific staining. Furthermore, embryos at this stage have a very nice internal control, given that the extraembryonic visceral endoderm is positive for Myc but the embryonic part of the visceral endoderm is negative for Myc expression. The absence of Myc detection in this tissue is then used to be sure of the absence of background staining. This is now indicated in the Figure and Figure legend.

3. In Fig. 3, the authors failed to obtain sub-cluster A and D homo deletion clones. Here more details are needed when performing the CRISPR experiment. If more than 100 clones were tested but still cannot get homo deletion, it will suggest an essential role of sub-cluster A and/or D in maintaining the Myc expression and cell proliferation. Since the hetero A or D deletion will likely has no phenotype, the authors should emphasize that the functions of sub-cluster A or D on Myc expression is nonconclusive.

The reviewer is absolutely correct. Both the absolute amount and the frequency of heterozygous deletions in these experiments indicated that homozygous deletions should be taking place, however they are not detected. While the simplest explanation would be that Myc expression is affected and required for ES cell viability, this is not the case, because Myc-deficient cells are perfectly viable (Scognamiglio, R. et al. Cell (2016) doi:10.1016/j.cell.2015.12.033, Alexandrova et al. Development (2016) 143 (1): 24–34 and our own unpublished results). Therefore, if homozygous clones are non-viable, the reason is not low Myc expression. The explanation of this phenomenon is therefore complex and beyond the scope of this manuscript.

4. In Fig. 3g', why are there few dots with high variance in naïve pluripotency data? Same in Fig. 4e', the data has few dots but high variance. Both experiments need to be repeated.

We thank the reviewer for this advice. Unfortunately, this condition produces results with a high variability. In revising the data we noticed some mistakes during data normalization and the wrong inclusion of some control measurements on GFP-Myc clones that were done in parallel experiments and should not have been included. We have now corrected this in figure 3, Figure 4 and Supplementary Figure 7. The results are now more consistent and the main conclusions about the activity of the genomic regions analyzed have not been modified by this reanalysis. The novelty is that cluster B deletion, which before showed only a tendency to elevated Myc levels in SR+LIF and N2B27+FA conditions, now shows significance. This is an interesting result given that the B region includes the Pvt1 promoter. This aspect is discussed below in response to reviewer 3 and has been commented in the manuscript.

5. In Supplemental Figure 4, please show WB data with both 65KD and 95KD region.

We now show the complete WB lanes.

Minor points:

1. Fig. 2h", the h" cannot be seen in the panel.

Corrected

2. “We also analyzed FP-MYC expression levels”. What does FP-MYC mean?

It means either GFP-Myc or TFP-Myc, depending on the genotype. We have now made this explicit.

3. The authors are advised to avoid using unconventional abbreviation words, such as epiLCs, epiSCs (should have capital "E"), CHIP (should be "ChIP"), and transcriptional start (should be "transcriptional start site").

Corrected, thank you

Reviewer #3 (Remarks to the Author):

In this manuscript, Li-Bao et al. interrogate the enhancer driven transcriptional regulation of the well-studied MYC gene in several different contexts including, two different pluripotent states in vitro and during early development in vivo. By knocking-in a BAC clone carrying 250kb around a MYC-TFP reporter gene and comparing the expression of this transgene to the endogenous MYC-GFP during development, they argue that this 250kb region is sufficient to recapitulate MYC regulation in early pluripotent and multipotent cells. In contrast, knock-in MYC-TFP reporter diverges from the endogenous MYC-GFP expression in later embryonic tissues, suggesting that additional regulatory elements beyond the 250kb region are required for proper regulation later in development. Then the researchers focus on individual candidate enhancers within the 250kb window, and identify two regions that when deleted show cell-state specific enhancer contribution to MYC expression. One of these regions has also been deleted in developing embryos, providing more robust evidence of its effect. The researchers use the logic that MYC expression is a readout for competitive fitness for selection of the "best" developmental cells. In support to this notion, deletions of specific MYC enhancers result into a negative selection, where cells that carry the deletion are outcompeted from the population.

Overall, the work is nicely presented and the data well-organized. The experiments done provide clear evidence that two cell-type specific enhancers (one for the naïve and one for the formative state) play a role in MYC regulation. However, there are several weaknesses on the narrative, the description and the justification of their approach and the conclusions that they draw from their data that need to be addressed. Below are my main concerns and suggestions:

We thank the reviewer for the comments and hope to have addressed the requests

Main Concerns:

- Selection and characterization of the candidate enhancer regions.
 - o Besides the genomic tracks of some ChIP-seq data in Figure 1i and the claim for "meta-analysis", there is no clear explanation of the criteria used for the selection of top candidate enhancers. Is it the relative strength of each of these signals? Any prioritization based on the type of TFs bound? As a reader I assumed that there was some integration of the deletions shown in 1A with the ChIP tracks shown in 1i. However, the deletions/perturbations shown in A are so large I am not sure what they add to the selection criteria.

We focused on regions showing high H3K27Ac and p300 signal in at least one of these three cell types: ESCs, EpiSCs and EpiLC, with preference for those that show a dynamic profile between these cell types, potentially indicating the presence of enhancers that regulate Myc expression in the different phases of pluripotency. We have now added this explanation to the main text.

- o The authors also present the MYC TAD that might eliminate interactions beyond its boundaries, they do not dig deeper into the actual insulation or relative contact frequencies with the putative enhancers. Given that a large number of high-resolution 3D assays (e.g micro-C, promoter Capture or HiChIP) are publically available both in mouse ESCs – and even EpiLCs (Capture-C), it is a wasted opportunity not to utilize these data to associate/predict regulatory impact in a cell-type specific manner. This would strengthen the novelty and relevance of the study.

We thank the reviewer for this suggestion. We have now included 10-kb resolution hi-C data in Supplementary Figure 1 and PLAC-seq data for H3K4me3, H3K27Ac and Pol2 in Figure 6. The data reinforce the idea that the BAC region contains ESC enhancers and that enhancers 2 and 7-3 are engaged in active interactions with the Myc promoter.

- o Moreover, the bioinformatics analysis of the candidate enhancers is very limited, simply listing the potentially relevant TFs that are bound there. Are these regions more enriched in binding or combinatorial bindings compared to other putative regulatory regions within the TAD? An easy follow up, would be to introduce point mutations or deletions of specific binding motifs to corroborate the relevance of one or more TF binding sites on the enhancer activity.

Please see the response to Reviewer 1 about the same question. We have now studied further the identity and roles of the putative transcription factors involved in regulating the identified enhancers. We have also extended the ChIPseq meta-analysis to a broader region around these enhancers. These new analyses and results are included in the new Figure 6 and Supplementary Figure 9.

In addition, as mentioned above, we generated mice with enhancer 2 deleted, which showed specific reduction of Myc expression in the naive epiblast (Figure 5 and Supplementary Figure 9).

- The establishment of a TFP-MYC transgene is a potentially useful tool that provided interesting insights, but with unclear relevance for this manuscript. Their experiments indeed support that the BAC clone likely contains all critical regulatory elements for MYC regulation in early development -although an actual quantitation and comparison of the MYC-TFP and endogenous MYC-GFP expressions is needed in the main figure (listing number of embryos and cells quantified). In contrast, their conclusion that regulatory elements required for later stages are likely located further away is not well-supported. The fact that the BAC is out of context might have some suppressive effects on the transgene, even if all important cell-type-specific enhancers are present. Another comment regarding the MYC-TFP experiment is that it is unclear how the introduction of this tool so early in the manuscript contributes to the follow up experiments shown after. Does the characterization of this reporter tool fit into the overall scope of the rest of the manuscript? What did the characterization of the MYC-TFP add to the overall criteria for choosing enhancer regions for deletion? It seems that the selection happened already based on Figure 1i.

Regarding the relevance of the use of the BAC, we think its characterization and its introduction early in the manuscript is very important and defined all subsequent work in the manuscript. While the initial large deletion and inversion analysis approximately delimits the BAC region, it only showed which regions are not needed in isolation for Myc expression, but it did not show which regions are sufficient for Myc expression in the early embryo and the BAC results were essential for this. Also, without the BAC, we could not exclude that combined redundant actions of remote regulatory regions would be responsible for Myc expression in the early embryo. The confirmation that the BAC contains all regulatory elements that provide the precise expression pattern of Myc in early mouse embryos was therefore an essential step for the further characterization of enhancers within the BAC region. The fact that we show the candidate enhancers in the first Figure is only for avoiding redundancies between the Figures, but it does not reflect the sequence of decisions during the project.

Regarding the quantitative aspects of the expression patterns in embryos, the quantification of the absolute levels of BAC-TFP versus Myc-GFP would not be informative, given that the detectability of the fluorescence from each reporter protein is different, however, the correlation studies performed in ES cells in different culture conditions show that not only the expression patterns but the cellular dynamic regulation of the endogenous GFP-Myc and the BAC-TFPMyc are similar. We therefore decided to perform a qualitative assessment of the expression pattern in embryos; i.e., to assess the similarity of the distribution in different tissues and embryonic structures of TFP-Myc versus GFP-Myc. We concluded that the qualitative distribution of both proteins is similar in early embryos.

Finally, the reviewer is right in that we cannot completely exclude the possibility that the BAC sequences contain enhancers for the regulation of Myc at later stages that would be suppressed by the insertion site influence. However, we think this is an unlikely possibility. On one side, BAC transgenesis is known for the reliability in reproducing endogenous expression patterns (for a review see for example *Transgenic Res.* 2001 Apr;10(2):83-103.doi: 10.1023/a:1008918913249.). On the other side, all embryonic expression patterns are fully conserved and reproduce endogenous dynamics and tissue-specific expression, however expression sharply shuts down when cells initiate differentiation. This observation shows that the BAC sequences are nicely isolated from the influence of the insertion site for the early embryonic expression and it would be difficult that it transits rapidly to a complete exposure to the insertion environment and complete suppression of the later enhancers. In addition, the notion that later embryonic enhancers lie outside the region characterized here is also supported by the previous characterization of the MNE and the BENC. Nonetheless, we have now rephrased these statements to make clear that we cannot completely exclude this possibility.

- The claim that MYC-enhancer deletions reduce fitness is very bold in context of the experiments done. Cell growth and differentiation experiments were done to show that upon MYC-enhancer deletion some cells are now less "fit" than others. The authors relate this process to selection of the best pluripotent cells in the embryo. However, all these experiments could simply indicate that manipulating MYC has an overall effect on proliferation, which is expected. Do they claim something beyond the obvious? Maybe it would be more convincing to demonstrate "fitness" in a developmental context. Would they contribute to embryos more?

We are confused by this comment. We do not refer to fitness selection or selection of the best pluripotent cells in the embryo when concluding about the experiments presented here. Fitness and cell selection is only mentioned in the introduction when citing previous work. From our experiments, we conclude that ES cells in which the identified enhancers are mutated are less competitive in co-culture and we think this is a reasonable description of the outcome of the experiments reported. The experiments in Figure 7 show that the variation produced in Myc

levels by the manipulations presented here have functional consequences for the cells, however the main topic of this manuscript is not cell competition but the identification of the Myc regulatory regions that control its expression in mammalian pluripotent cells. For these reasons, we think a thorough study of how the different mutations described here affect cell competition *in vivo* is more appropriate for a future study. Nonetheless, based on previous studies (Clavería et al Nature 2013; Sancho et al., Dev cell 2013; Diaz-Diaz et al, Dev Cell 2017, Hashimoto and Sasaki Dev Cell 2019) the prediction is that cells carrying enhancer mutations that reduce Myc expression would show reduced *in vivo* competitiveness.

Minor Concerns:

o Authors describe the SR+LIF condition as “mixed pluripotency”, by which I assume they mean a mixture of naïve and formative pluripotent cells. Where do they base this description? The paper they cite is not relevant. The reviewer is correct; we characterized this in our previous work Diaz-Diaz et al Dev Cell 2017. This reference is now included

o How do any of the reported deletions affect PVT1 gene? There are various reports showing competition between MYC and PVT1. It is only mentioned in the discussion “These results suggest that neither the long-noncoding RNA encoded by Pvt1, nor its transcriptional activity regulate Myc expression in ES cells”, but I could not find the data supporting this claim.

Pvt1 is a 226 Kb transcript thought to cooperate with Myc in tumorigenesis, as shown by several reports (referenced in the manuscript).

Another report, also cited in the manuscript, shows the interesting phenomenon mentioned by the reviewer of competition between the Pvt1 and Myc promoters for transcription (Sho S.W. et al, Cell 2018).

The data that support the statement that Pvt1 does not promote Myc expression in the early embryo are:

- 1.- The BAC reproduces Myc expression levels and dynamics in the early embryo and embryonic stem cells while it lacks 73 kb of the 3' of the transcriptional unit.
- 2.- The B subcluster deletion eliminates the Pvt1 promoter and 6.2 Kb of its 5' region and increases Myc expression

We think these results strongly suggest that, opposite to what has been reported in Tumors, the Pvt1 transcript does not promote Myc expression in ES cells

Interestingly, the results obtained with the B deletion indeed suggest that the competition between promoters takes place in ES cells. This is now explicit in the Discussion section.

o All imaging data should be quantified listing the numbers of embryos and cells used.

We have now included this information in the Figure legend, Figures or Source Data made available through Figshare (<https://figshare.com/s/8c9aa8447a01db9f98bd>)

o In Figure F4G the authors do not describe the image and the TF enrichment adequately. What do they consider binding and what do these lines represent (# of reads)? Do they consider FOXD3 as a binding factor at enhancer-2 based on 2 lines?

This is a meta-analysis. Each line represents a peak identified in an independent ChIPseq analysis. Two lines therefore indicate two independent ChIP-seq analyses in the literature. We have now explained this better in the Figure legend (Now figure 6) and changed the representation for ease of visualization.

o 4C analysis is missing since there is no mention of the algorithms and steps used to generate supplementary figure 2a and 2b.

We have now included this information in the Methods section. In addition, the data are available from GEO and a token is provided in the manuscript for the reviewer to access the data

o In page 8 it is written: “While we could not obtain sub-clusters A and D homozygous deletions, sub-cluster C deletion in heterozygosity provokes detectable MYC level reduction in SR+LIF and F/A conditions (Supplementary Figure 7a-e), suggesting that if sub-clusters A and D contained major enhancer activity in these culture conditions, it would have been detected.” A better explanation should be provided.

The reviewer is correct on this point also raised by reviewer 1; we have now clarified this question in the main text

o Figure 5 is presented before figure 4 in the manuscript and somehow this complicates the flow.
We have now added a new figure and reorganized the Figures for better flow

REVIEWER COMMENTS

Reviewer #1 (Remarks to the Author):

In this revised manuscript, the authors made extensive revisions as the reviewers' suggestions. Especially for this reviewer, they performed fine deletion analyses of the enhancers to reveal the roles of the transcription factors. This made me to consider the revised version as a candidate for publication. Overall, the revised version looks fine and only few points will be required additional revisions.

1. In abstract, the authors stated that the transcription factor MYC plays essential roles in pluripotent stem cells, including the promotion of somatic cell reprogramming to pluripotency and the regulation of cell competition and embryonic diapause. This is obvious overstatement because Myc-null ES cells continue self-renewal (Davis et al, Genes Dev, 1993) and Myc is not essential for reprogramming.
2. In Figure 5, the authors performed the competitive assay using various mutant cell lines. They should confirm the proliferation speed of each cell line in the clonal culture condition.
3. The discrepancy between the impacts of enhancer 7-3 and the small deletion in it is mysterious. Was only single ESC line analyzed? Analysis of multiple clones will be recommended to exclude the possibility of off-target mutation by Crispr/Cas9.

Reviewer #2 (Remarks to the Author):

In this revised manuscript, the authors have addressed most of my concerns raised in the initial review, except for point #3. The authors argued that it is out of the scope of the current manuscript, but I strongly believe that it is relevant and should at least be discussed why sub-cluster A and D homo deletion clones cannot be obtained. I did not see such an explanation in the current Discussion section.

Also, there are still a few minor issues:

1. Supplementary Fig. 1c was mentioned, but nowhere to be seen.
2. On page 10, "NME" was mentioned, but what is it?
3. Page 14, lines 1-2: It is a broken sentence.

Reviewer #3 (Remarks to the Author):

The authors performed additional experiments to strengthen the functional characterization of selected Myc enhancers both in vitro and in vivo. They also improved the analysis and provided additional clarifications in the text. Altogether, I believe that they have addressed most of the reviewers' comments in a satisfactory manner. My only remaining comments have to do with figure legends either completely missing (e.g. the legends for Figure panels 4g-k are completely missing) or incompletely/inaccurately described (e.g. in Fig. 6e they mention that each point represents a biological replicate, but there are no points illustrated- only a bar graph). Overall, a careful proofreading of all figure legends and accurate description of the results, including the numbers and types of replicates (technical or independent clones for example?), the number of independent experiments and the number of nuclei should be clearly stated.

REVIEWER COMMENTS

Reviewer #1 (Remarks to the Author):

In this revised manuscript, the authors made extensive revisions as the reviewers' suggestions. Especially for this reviewer, they performed fine deletion analyses of the enhancers to reveal the roles of the transcription factors. This made me to consider the revised version as a candidate for publication. Overall, the revised version looks fine and only few points will be required additional revisions.

1. In abstract, the authors stated that the transcription factor MYC plays essential roles in pluripotent stem cells, including the promotion of somatic cell reprogramming to pluripotency and the regulation of cell competition and embryonic diapause. This is obvious overstatement because Myc-null ES cells continue self-renewal (Davis et al, Genes Dev, 1993) and Myc is not essential for reprogramming.

We thank the reviewer for noticing this. In fact the sole mutation of Myc does not impair ES cell ability to renew or differentiate. This was shown as well in Scognamiglio et al, Cell 2016 and in Alexandrova et al. Development 2016. We have now rephrased the abstract for a more accurate description of Myc roles, which are redundant with Mycn for diapause induction (Scognamiglio et al Cell 2016) and dispensable but cooperative for reprogramming. The new sentence in the abstract states: "The transcription factor MYC plays various roles in pluripotent stem cells, including the promotion of somatic cell reprogramming to pluripotency, the regulation of cell competition and the control of embryonic diapause."

2. In Figure 5, the authors performed the competitive assay using various mutant cell lines. They should confirm the proliferation speed of each cell line in the clonal culture condition.

We guess the reviewer suggests to study the expansion of the populations when cultured in isolation. Indeed, this is regular practice when studying cell competition. We now provide data in the new Supplementary Figure 12 of the expansion of the Myc-GFP and the sub-cluster C-deleted Myc-GFP cells in comparison with the WT-Tomato cells when grown independently. The results show that sub-cluster C-deleted cells grow at equal rates than WT cells when grown in isolation in either the SR+LIF or the N2B27+FA conditions. In addition, in the same Figure, we provide evidence of cell death rate increase in sub-cluster C-deleted ESCs upon co-culture with WT cells. The results agree with the previously described dynamic of Myc-mediated cell competition, which relies on non-autonomous cell death and not in differential cell cycle activity (Sancho et al Dev Cell 2013; Claveria et al Nature 2013; Hashimoto and Sasaki Dev Cell 2019). These results are included as well in the new Supplementary Figure 12 and described in the Results section.

3. The discrepancy between the impacts of enhancer 7-3 and the small deletion in it is mysterious. Was only single ESC line analyzed? Analysis of multiple clones will be recommended to exclude the possibility of off-target mutation by Crispr/Cas9.

We understand and share the concerns of the reviewer on this issue, however, this would only be solved by obtaining the same or a larger deletion (>31 bp) in homozygosity in an independent clone; otherwise, any difference observed could be attributed to the different mutation. Large deletions are however rare using single crRNA. We have isolated and screened 15 independent clones mutated with the crRNA targeted to the Otx2 region but failed to identify any suitable second clone for this analysis. For this mutagenesis, we used the "Alt-R S.p. HiFi Cas9 Nuclease V3", which shows reduced off-target cutting events compared with other Cas9 enzymes, with similar high on-target activity. To check for off-target alterations, we then amplified and sequenced the three top predicted off-target sites for each cell clone studied and found neither alterations in the PCR bands amplified nor in the DNA sequence of the Otx2 clone or in any other mutant clone obtained. These results are shown in the new Supplementary Figure 9 and explained in the Results section.

Reviewer #2 (Remarks to the Author):

In this revised manuscript, the authors have addressed most of my concerns raised in the initial review, except for point #3. The authors argued that it is out of the scope of the current manuscript, but I strongly believe that it is relevant and should at least be discussed why sub-cluster A and D homo deletion clones cannot be obtained. I did not see such an explanation in the current Discussion section.

We thank the reviewer for this comment and have now included this issue in the Discussion section

Also, there are still a few minor issues:

1. Supplementary Fig. 1c was mentioned, but nowhere to be seen.

Thank you for noticing. The right call is to Supplementary Figure 1c

2. On page 10, "NME" was mentioned, but what is it?

Thank you for noticing. This was a typo, now corrected to NPE (naive pluripotency enhancer)

3. Page 14, lines 1-2: It is a broken sentence.

Thank you for noticing. This is now fixed

Reviewer #3 (Remarks to the Author):

The authors performed additional experiments to strengthen the functional characterization of selected Myc enhancers both in vitro and in vivo. They also improved the analysis and provided additional clarifications in the text. Altogether, I believe that they have addressed most of the reviewers' comments in a satisfactory manner. My only remaining comments have to do with figure legends either completely missing (e.g the legends for Figure panels 4g-k are completely missing) or incompletely/inaccurately described (e.g in Fig.6e they mention that each point represents a biological replicate, but there are no points illustrated- only a bar graph). Overall, a careful proofreading of all figure legends and accurate description of the results, including the numbers and types of replicates (technical or independent clones for example?), the number of independent experiments and the number of nuclei should be clearly stated.

Thank you for the comments. The original Figure 4 was splitted into Figs. 4 and 6 in the last submitted version. Unfortunately in making the pdf, the original Figure 4 was included instead of the new reduced Figure 4. Therefore some panels were duplicated between Figure 4 and 6 but not the legends. This is now fixed.

We have gone through all Figure legends and described accurately the number of replicates and their nature. All replicates were biological (e. g. different culture dishes or embryos). The mention to "technical replicates" was a mistake and this is now corrected.

In addition, the source data are available from Figshare, where all the original data can be accessed. For Cytometry analyses, the N is available from the source data in Figshare and this has been now mentioned in all Figure legends when applicable.

REVIEWERS' COMMENTS

Reviewer #1 (Remarks to the Author):

In this second revision, the authors gave proper answers to this reviewer's questions, so now I agree with the publication of the present version of the manuscript.